



# A comparative isotopic study of the biogeochemical cycle of carbon in modern stratified lakes: the hidden role of DOC

Robin Havas[a,*], Christophe Thomazo[a,b], Miguel Iniesto[c], Didier Jézéquel[d], David Moreira[c], Rosaluz Tavera[e], Jeanne Caumartin[f], Elodie Muller[f], Purificación López-García[c], Karim Benzerara[f]

[a] Biogéosciences, CNRS, Université de Bourgogne Franche-Comté, 21 000 Dijon, France

[b] Institut Universitaire de France, 75005 Paris, France

[c] Ecologie Systématique Evolution, CNRS, Université Paris-Saclay, AgroParisTech, 91190 Gif-sur-Yvette, France

[d] IPGP, CNRS, Université de Paris, 75005 Paris, and UMR CARRTEL, INRAE & USMB, France

[e] Departamento de Ecología y Recursos Naturales, Universidad Nacional Autónoma de México, México

[f] Sorbonne Université, Muséum National d'Histoire Naturelle, CNRS, Institut de Minéralogie, de Physique des Matériaux et de Cosmochimie (IMPMC), 75005 Paris, France.

[*] *Correspondence to*: Robin Havas (robin.havas@gmail.com)

*Keywords: Carbon cycle; isotopic fractionation; DOC; Precambrian analogs*



**Abstract.** The carbon cycle is central to the evolution of biogeochemical processes at the surface of the Earth. In the ocean, which has been redox-stratified through most of the Earth's history, the dissolved organic carbon (DOC) reservoir holds a critical role in these processes because of its large size and involvement in many biogeochemical reactions. However, it is rarely measured and examined in modern stratified analogs and yet commonly invoked in past C cycle studies. Here, we characterized the C cycles of four redox-stratified alkaline crater lakes from Mexico. For this purpose, we analyzed the concentrations and isotopic compositions of DOC together with dissolved inorganic and particulate organic C (DIC and POC). In parallel we measured physico-chemical parameters of the water columns and surficial bottom sediments. The four lakes have high DOC concentrations (from ~ 15 to 160 times the amount of POC, averaging $2 \pm 4$ mM; 1SD, n=28) with an important variability between and within the lakes. All lakes exhibit prominent DOC peaks (up to 21 mM), found in the oxic and/or anoxic zones. $\delta^{13}C_{DOC}$ signatures also span a broad range of values from -29.3 to -8.7 ‰ (with as much as 12.5 ‰ variation within a single lake), while $\delta^{13}C_{POC}$ and $\delta^{13}C_{DIC}$ varied from -29.0 to -23.5 ‰ and -4.1 to +2.0 ‰, respectively. The DOC peaks in the water columns and associated isotopic variability seem mostly related to oxygenic and/or anoxygenic primary productivity through the release of excess fixed C in three of the lakes (Atexcac, La Preciosa and La Alberca de los Espinos). By contrast, the variability of [DOC] and $\delta^{13}C_{DOC}$ in Lake Alchichica could be mainly explained by partial degradation and accumulation in anoxic waters. Overall, DOC records metabolic reactions that would not have been clearly detected if only DIC and POC reservoirs had been analyzed. For example, DOC analyses evidence an active DIC-uptake and use of a DIC-concentrating mechanism by part of the photosynthetic plankton. Despite the prominent role of DOC in the C cycle of these lakes, variations of [DOC]/$\delta^{13}C_{DOC}$ and associated reactions are not reflected in the sedimentary organic carbon record, hence calling for special care when considering sediments as reliable archives of metabolic activities in stratified water columns. Overall, this study brings to light the need of further investigating the role of DOC in the C cycles of modern stratified analogs.





## 1. INTRODUCTION

The carbon cycle and biogeochemical conditions prevailing at the surface of the Earth are intimately bound through combined biological and geological processes and have evolved together throughout the Earth's history. Accordingly, the analysis of carbon isotopes from the rock record has been used to reconstruct the evolution of the biosphere and oxygenation of the Earth's (e.g. Hayes et al., 1989; Karhu and Holland, 1996; Schidlowski, 2001; Bekker et al., 2008). Because the oceans have been redox-stratified throughout most of the Earth's history (Lyons et al., 2014; Havig et al., 2015; Satkoski et al., 2015), processes affecting the C cycle were likely different from those occurring in most modern, well oxygenated environments. This impacts from the diversity and relative abundance of microbial carbon and energy metabolism (e.g. Paneth and O'Leary, 1985; Hessen and Anderson, 2008; Wang et al., 2016; Iñiguez et al., 2020), to larger ecological interactions (e.g. Jiao et al., 2010; Close and Henderson, 2020; Klawonn et al., 2021) and global C dynamics (e.g. Ridgwell and Arndt, 2015; Ussiri and Lal, 2017). Nonetheless, some modern stratified analogs (anoxic at depth) of early oceans exist, and need to be characterized in order to better understand the C cycle in ancient redox-stratified systems and how it was recorded by the sedimentary archives (e.g. Lehmann et al., 2004; Posth et al., 2017; Fulton et al., 2018). To this end, a number of recent studies investigated the C cycle of modern stratified water columns (e.g. Crowe et al., 2011; Kuntz et al., 2015; Camacho et al., 2017; Posth et al., 2017; Schiff et al., 2017; Havig et al., 2018; Cadeau et al., 2020; Saini et al., 2021; Petrash et al., 2022), with the clear advantage that many of their bio-geo-physico-chemical parameters could be directly measured, together with the main C reservoirs.

Yet, very few studies on redox-stratified analogs included the analysis of dissolved organic carbon (DOC) and even fewer measured DOC stable isotope data (Havig et al., 2018), despite the fact that it is a major component of marine and fresh waters (e.g. Ridgwell and Arndt, 2015; Brailsford, 2019). Indeed, DOC generally represents the majority of freshwater organic matter (Kaplan et al., 2008; Brailsford, 2019), while the size of oceanic DOC equals the total amount of atmospheric carbon (Jiao et al., 2010; Thornton, 2014). DOC is (i) at the base of many trophic chains (Bade et al., 2007; Hessen and Anderson, 2008; Jiao et al., 2010; Thornton, 2014), (ii) key in physiological and ecological equilibria (Hessen and Anderson, 2008) and (iii) has a critical role as a long-term C storage reservoir for climate change (Jiao et al., 2010; Hansell, 2013; Thornton, 2014; Ridgwell and Arndt, 2015). Although the contribution of DOC reservoirs to the past and modern Earth's global climate and biogeochemical cycles is not properly constrained (Jiao et al., 2010; Dittmar, 2015), it has been used to explain some perturbations of the C cycle recorded in sedimentary archives (e.g. Rothman et al., 2003; Fike et al., 2006; Sexton et al., 2011; Ridgwell and Arndt, 2015).

Here, we propose to describe the C cycle of four modern redox-stratified alkaline crater lakes, located in the trans-Mexican volcanic belt (Ferrari et al., 2012). They relate to similar geological contexts and climates but have distinct solution chemistries – aligning along an alkalinity/salinity gradient (Zeyen et al., 2021) – as well as distinct planktonic communities (Iniesto et al., in press). Moreover, they harbor various types of microbialites (Gérard et al., 2013; Saghaï et al., 2016; Iniesto et al., 2021a, 2021b; Zeyen et al., 2021). We measured the concentrations and isotopic compositions of C-containing phases throughout the stratified water column of the lakes, including DOC, dissolved inorganic carbon (DIC) and particulate organic carbon (POC). In parallel, depth profiles of several physico-chemical parameters as well as trace and major elements concentrations were measured allowing to pinpoint the main occurring biogeochemical reactions and connect them with specific C isotopes signatures. Last,



surficial sediments (~ 10 cm) at the bottom of the lakes were also characterized in order to further constrain the
main geochemical reactions taking place in the lower water columns and infer possible exchanges between the
sediment and water column reservoirs.
Investigations of the C cycle in Precambrian analogs usually focus on a single environment instead of integrating
views from several systems (e.g. Camacho et al., 2017; Schiff et al., 2017). Here, the inter-comparison via the
same methodology of four redox-stratified lakes of the same type (tropical alkaline volcanic crater-lakes) but with
distinct solution chemistries and microbial diversities (Zeyen et al., 2021; Iniesto et al., in press), allows to assess
the effects of specific physico-chemical and biological parameters on the C cycle. Thus, we present the main
biogeochemical reactions occurring in the water columns (e.g. oxygenic/anoxygenic photosynthesis or
methanogenesis) and how they are recorded (or not) in surficial sediments. Then, we shed a new light on the
microbial cycling of DOC and how the analysis of its isotopes can provide deeper insights into microbial processes
and overall C cycle dynamics in stratified water columns. Finally, we discuss the possible implications of DOC
for paleoclimate reconstruction and the interpretation of the sedimentary C isotopes record.

## 2.   SETTING / CONTEXT


### 2.1. Geology


The four lakes studied here are volcanic maars formed after phreatic, magmatic and phreatomagmatic explosions,
in relation with volcanic activity in the Trans-Mexican-Volcanic Belt (TMVB, Fig. 1). TMVB originates from the
subduction of the Rivera and Cocos plates beneath the North America plate, resulting in a long and wide (~1000
and 90-230 km, respectively) Neogene volcanic arc spreading across central Mexico (Ferrari et al., 2012). The
TMVB harbors a large variety of monogenetic scoria cones and phreatomagmatic vents (maars and tuff-cones) as
well as stratovolcanoes, calderas and domes (Carrasco-Núñez et al., 2007; Ferrari et al., 2012; Siebe et al., 2014).
Maar crater formation usually occurs when ascending magmas meet water-saturated substrates, leading to
successive explosions downward within and excavation of older units (Lorenz, 1986; Carrasco-Núñez et al., 2007;
Siebe et al., 2012; Chako Tchamabé et al., 2020).
Three of the studied lakes (Alchichica, Atexcac and La Preciosa) are located in a restricted area (~ 50 km²) of the
Serdan-Oriental Basin (SOB) in the easternmost part of the TMVB (Fig. 1). The SOB is a closed intra-montane
basin at high altitude (~2300 m), surrounded by the Los Humeros caldera in the north and Cofre de Perote-
Citlatépel volcanic range in the east. The basement is mainly composed of folded and faulted Cretaceous
limestones and shales, covered by andesitic to basaltic lava flows (Carrasco-Núñez et al., 2007; Armienta et al.,
2008; Chako Tchamabé et al., 2020). The formations of Alchichica and Atexcac craters were dated back to ~ 6-13
± 5-6 and 330 ± 80 ka, respectively (Table 1; Carrasco-Núñez et al., 2007; Chako Tchamabé et al., 2020). The age
of lake La Preciosa is not known. The fourth lake, La Alberca de los Espinos, is located at the margin of the Zacapu
tectonic lacustrine basin in the Michoacán-Guanajuato Volcanic Field (MGVF) in the western-central part of the
TMVB (Fig. 1). It lies at about 1985 m, mainly on andesitic basement rocks and was dated at ~25 ± 2 ka years
(Siebe et al., 2012, 2014).

Biogeosciences Open Access
Discussions
EGU

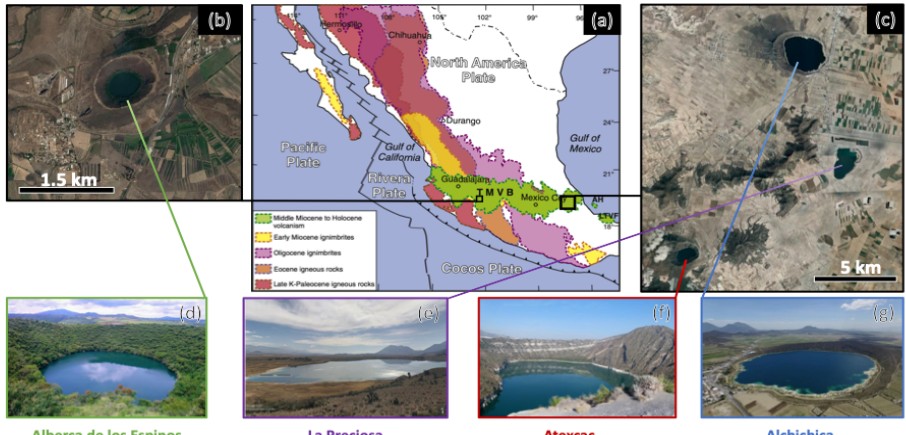

Figure 1. Geographical location and photographs of the studied crater lakes. (a) Geological map from
Ferrari et al. (2012) representing the location of the four studied lakes in the trans-Mexican volcanic
belt (TMVB). (b), (c) Close up © Google Earth views of lake Alberca de los Espinos and the Serdan-
Oriental Basin (SOB), respectively. (d-g) Pictures of the four studied lakes (d from © Google Image
['enamoratedemexicowebsite'], e from © Google Earth street view, and g from © 'Agencia Es Imagen').

**2.2. Climate and limnology**
Due to their geographical proximity from each other, lakes from the SOB experience a similar temperate to semi-
arid climate (Armienta et al., 2008; Sigala et al., 2017). The present climate of the SOB is dominated by dry
conditions with evaporation fluxes higher than precipitation fluxes in Lake Alchichica for example (~ 1686 *vs*
392 mm/an; Alcocer, 2021). In Alchichica, Atexcac and La Preciosa, this trend is reflected by a drop in their water
level evidenced by the emersion of microbialite deposits in these lakes (Fig. S1; Zeyen et al., 2021). This
evaporation-dominated climate strongly contributes to the achievement of relatively high lake salinities from 1.2
to 7.9 psu, ranging from sub- to hyposaline. In comparison, Alberca's climate is temperate to semi-humid and it is
a freshwater lake (0.6 psu, Rendon-Lopez, 2008; Sigala et al., 2017).
The four lakes are warm monomictic, i.e., they are stratified throughout most of the year (~ 9 months) and mix
once a year when the thermal stratification breaks down in the cold winter (Armienta et al., 2008). They are all
"closed lakes" located in an "endorheic" basin (Alcocer, 2021; Zeyen et al., 2021), meaning that they have no
inflow, outflow nor a connection to other basins through surficial waters such as streams. Overall, water input is
only supplied by precipitations, and groundwater inflow as evidenced and quantified for Lake Alchichica (Alcocer,
2021 and references therein).
Finally, the four lakes are alkaline (pH ~ 9) but distribute over a gradient of chemical compositions (including
alkalinity, salinity and Mg/Ca ratio) interpreted as reflecting varying concentration stages of an initial alkaline
dilute water (Table 1; Zeyen et al., 2021), evolving due to different climates (mostly between Alberca and lakes



from the SOB) and more generally, different hydrological regimes. Microbialite deposits are found in all four lakes
with an increasing abundance from lower to higher alkaline conditions (Zeyen et al., 2021).
Table 1. General information of the studied lakes. Abbreviations: TMVB: Trans-Mexican volcanic
belt; MGVF: Michoaćan-Guanajuato volcanic field; masl: meters above sea level. *NB*: Sampling in
May 2019 except for La Preciosa's sediments, sampled in May 2016.

| Lake | General location | Sampling location | Elevation |
|---|---|---|---|
| Alchichica | Serdan Oriental Basin, eastern TMVB | 19°24'51.5" N; 097°24'09.9" W | 2320 |
| Atexcac | Serdan Oriental Basin, eastern TMVB | 19°20'2.2'' N; 097°26'59.3'' W | 2360 |
| La Preciosa | Serdan Oriental Basin, eastern TMVB | 19°22'18.1'' N; 097°23'14.4'' W | 2330 |
| La Alberca de los Espinos | Zacapu Basin, MGVF, central TMVB | 19°54'23.9'' N; 101°46'07.8'' W | 1985 |


| Lake | Lake Basement | Age | Max Depth (m) | Alkalinity (mmoles/L) | Salinity (psu) | pH |
|---|---|---|---|---|---|---|
| Alchichica | limestone, basalts | 6-13 ± 5-6 ka | 63 | ~35 | 7.9 | 9.22 |
| Atexcac | limestone, andesites, basalts | 330 ± 80 ka | 39 | ~26 | 7.4 | 8.85 |
| La Preciosa | limestone, basalts | Pleistocene | 46 | ~13.5 | 1.15 | 9.01 |
| La Alberca de los Espinos | andesite xenoliths | 25 ± 2 ka | 30 | ~7 | 0.6 | 9.14 |



## 3. METHOD

### 3.1. Sample Collection

The sediment core from Lake La Preciosa was collected in May 2016. All other samples were collected in May
2019. The depth profiles of several physico-chemical parameters were measured in the water columns of the four
lakes using an YSI Exo 2 multi-parameter probe: temperature, pH, ORP, conductivity, $O_2$, chlorophyll a,
phycocyanin, and turbidity. Precisions for these measurements were 0.01 °C, 0.1 pH unit, 20 mV, 0.001 mS/cm,
0.1 mg/L, 0.01 µg/L, 0.01 µg/L and 2% FNU unit, respectively. The ORP signal was semi-calibrated and was used
to report relative variations over a depth profile. Measurements of the aforementioned parameters allowed to
pinpoint depths of interest for further chemical and isotopic analyses, notably around the chemoclines of the lakes.
Water samples were collected with a Niskin bottle. For analyses of dissolved inorganic and organic carbon (DIC,
DOC), major, minor and trace ions, between 1.5 and 5 L of lake water were filtered at 0.22 µm with Filtropur S
filters that were pre-rinsed with the lake water. Particulate matter was collected on pre-ashed and weighted glass
fiber filters (Whatman GF/F, 0.7 µm) and analyzed for particulate organic carbon (POC), major and trace elements.





Sediment cores were collected using a 90 mm Uwitec corer at the bottom of each lake's where the water column
was at its deepest (Table 1) and anoxic conditions prevail almost all year long. Cores measured between 20 and
85 cm in length. Slices of about 2-3 cm were cut. Interstitial pore water was drained out of the core slices using
Rhizons. Sediments were then fully dried in a laboratory anoxic $N_2$-filled glove box.

### 3.2. Dissolved inorganic carbon (DIC) concentration and isotope measurements

Twelve milliliters of 0.22-µm-filtered solutions were placed in hermetic Exetainer® tubes in order to avoid
exchange between DIC and atmospheric $CO_2$. DIC concentrations and isotopic compositions were determined at
the Institut de Physique du Globe de Paris (IPGP), using an Analytical Precision 2003 GC-IRMS, running under
He-continuous flow, and following the protocol described by Assayag et al. (2006). In short, a given volume of
water sample is taken out of the Exetainer® tube with a syringe, while the same volume of helium is introduced
in order to maintain a stable pressure and atmospheric-$CO_2$-free conditions within the sample tubes. The collected
sample is introduced in another Exetainer® tube that was pre-filled with a few drops of 100% phosphoric acid
($H_3PO_4$) and pre-flushed with He gas. Under acidic conditions, the DIC quantitatively converts to gaseous and
aqueous $CO_2$, which equilibrates overnight within the He filled head space of the tube. Quantification and isotopic
analyses of released gaseous $CO_2$ are then carried out by GC-IRMS using internal standards of known composition
that were prepared and analyzed via the same protocol. Each measurement represents an average of four injections
in the mass spectrometer. Chemical preparation and IRMS analysis were duplicated for all the samples. The $\delta^{13}C_{DIC}$
reproducibility calculated for the 65 samples was better than ±0.2 ‰, including internal and external
reproducibility. Standard deviation for [DIC] was 0.6 ± 0.9 mmoles/L on average.
Specific DIC speciation, i.e., $CO_{2(aq)}$, $HCO_3^-$ and $CO_3^{2-}$ activities, was computed using Phreeqc with the full
dissolved chemical composition of each sample as an input. It should be noted that these results are provided by
calculations of theoretical chemical equilibria and do not necessarily take into account local kinetic effects, which,
for example, could lead to local exhaustion of $CO_{2(aq)}$ where intense photosynthesis occurs. Additionally, dating
of the DIC was achieved by measuring its $^{14}C$ content and was performed by Beta Analytic laboratory, Miami,
USA.

### 3.3. Dissolved organic carbon (DOC) concentrations and isotope measurements

Samples of filtered solutions were first acidified to a pH of about 1-2 in order to degas all the DIC and preserve
the DOC only. DOC concentrations were measured with a Vario TOC at the Biogéosciences Laboratory calibrated
with a range of potassium hydrogen phthalate (Acros®) solutions. Before isotopic analyses, DOC concentration
of the samples was adjusted to match our international standards at 5 ppm (USGS 40 glutamic acid and USGS 62
caffeine). Isotopic compositions were measured at the Biogéosciences Laboratory using an IsoTOC (running under
He-continuous flow) coupled with an IsoPrime stable isotope ratio mass spectrometer (IRMS; Isoprime,
Manchester, UK). Samples were stirred with a magnetic bar and flushed with He before injection of 1 mL sample
aliquots (repeated 3 times). DOC is then transformed into gaseous $CO_2$ by combustion at about 850 °C,
quantitatively oxidized by copper oxides and separated from other combustion products in a reduction column and



water condensers. Finally, it is transferred to the IRMS via an open split device. In order to avoid a significant
memory effect between consecutive analyses, samples were separated by six aliquots of deionized water and their
first aliquot was discarded from the isotopic calculations. Average reproducibility of $\delta^{13}C_{DOC}$ on standards and
samples was 1 and 0.5 ‰ (1SD), respectively. Average reproducibility for sample [DOC] measurements was on
average 0.3 mM.

### 3.4. Particulate organic carbon and nitrogen (POC / PON)

Particulate organic matter from the lakes water columns was collected on GF/F quartz filters, ground in a ball mill
before and after decarbonation. Decarbonation was performed with 12N HCl vapors in a desiccator for 48 h.
Aliquots of dry decarbonated samples (25 - 70 mg) were weighed in tin capsules. POC and PON contents and
$\delta^{13}C_{POC}$ were determined at the Biogéosciences Laboratory using a Vario MICRO cube elemental analyzer
(Elementar, Hanau, Germany) coupled in continuous flow mode with an IsoPrime IRMS (Isoprime, Manchester,
UK). USGS 40 and IAEA 600 certified materials were used for calibration and showed a reproducibility better
than 0.15 ‰ for $\delta^{13}C$. External reproducibility based on triplicate analyses of samples (n=23) was 0.1 ‰ on average
for $\delta^{13}C_{POC}$ (1SD). External reproducibilities of POC and PON concentrations were on average 0.001 and
0.005 mmoles/L, respectively (i.e. 3 and 7 % of measured concentrations).

### 3.5. Geochemical characterizations of the sediments

Sedimentary organic carbon (SOC), sedimentary organic nitrogen (SON) and their isotopic compositions were
measured on carbonate-free residues of the first 12 cm of sediments, produced after overnight 1N HCl digestion.
Plant debris (mainly found in La Alberca and Atexcac) were picked upon initial sediment grinding in an agate
mortar and analyzed separately. Aliquots of dried decarbonated samples (~ 4-70 mg) were weighed in tin capsules.
SOC and SON contents and $\delta^{13}C$ were determined at the Biogéosciences Laboratory using a Vario MICRO cube
elemental analyzer (Elementar GmbH, Hanau, Germany) coupled in continuous flow mode with an IsoPrime
IRMS (Isoprime, Manchester, UK). USGS 40 and IAEA 600 certified materials were used for calibration and had
reproducibility better than 0.2 ‰ for $\delta^{13}C_{SOC}$. Sample analyses (n=67) were at least duplicated and showed an
average external reproducibility of 0.1 ‰ for $\delta^{13}C$ (1SD). External reproducibilities for SOC and SON contents
were 0.1 and 0.03 wt. %, respectively.
Carbon isotopic compositions of carbonates from the bottom sediments in La Alberca were analyzed at the
Biogéosciences Laboratory using a ThermoScientific™ Delta V Plus™ IRMS coupled with a Kiel VI carbonate
preparation device. External reproducibility was assessed by multiple measurements of NBS19 standard and was
better than ± 0.1 ‰ (2σ).
Solid sulfides minerals concentrations were determined on dry bulk sediments in Lake La Alberca after a wet
chemical extraction using a boiling acidic Cr(II)-solution as detailed in Gröger et al. (2009).

### 3.6. Major and trace elements concentrations

Dissolved and particulate matter elemental compositions were measured at the Pôle Spectrométrie Océan
(Plouzané, France) by inductively coupled plasma atomic-mass spectroscopy (ICP-AES, Horiba Jobin) for major
elements and by high resolution-ICP-mass spectrometry using an Element XR (HR-ICP-MS, Thermo Fisher
Scientific) for trace elements. Major element measurement reproducibility based on internal multi-elemental
solution was better than 5%. Trace elements were analyzed by a standard-sample bracketing method and calibrated
with a multi-elemental solution. Analytical precision for trace elements was generally better than 5%. Dissolved
sulfates/chloride and ammonium concentrations ($NH_4^+$) were determined at the IPGP by chromatography and by
continuous flow colorimetric analysis, respectively, with an uncertainty lower than 5%.

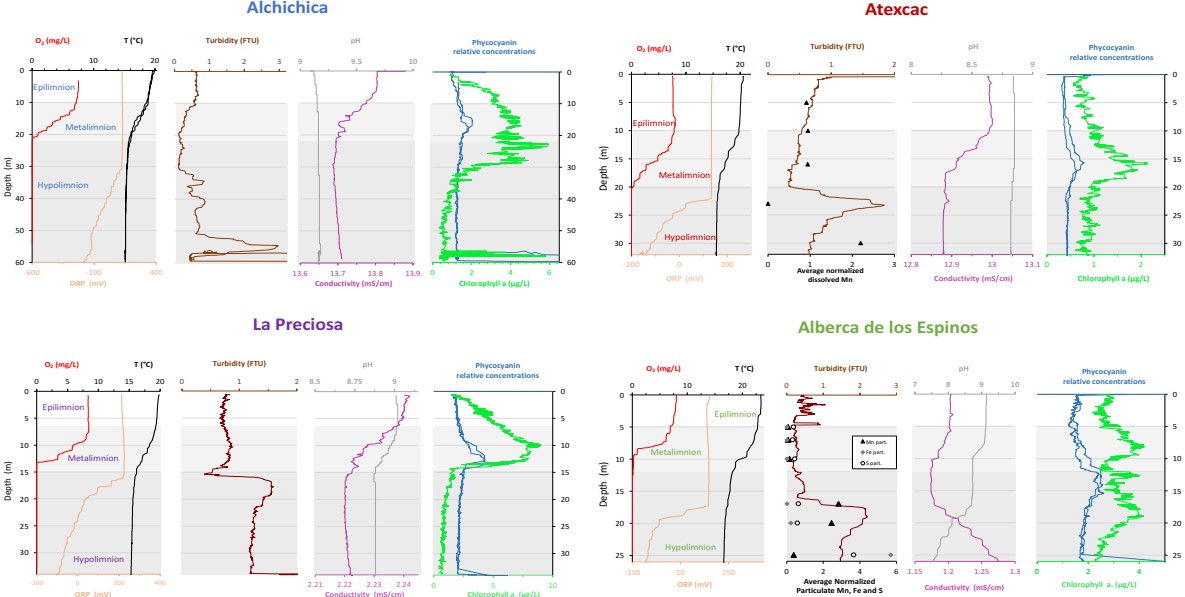

Figure 2. Physico-chemical parameters depth profiles of Alchichica, Atexcac, La Preciosa and Alberca de los
Espinos including: dissolved oxygen concentrations (mg/L), water temperature (°C), oxidation-reduction
potential (ORP, mV), turbidity (Formazin Turbidity Unit), pH, conductivity (mS/cm), phycocyanin and chlorophyll
a. pigments (µg/L). Absolute values for phycocyanin concentrations were not determined; only relative variations
are represented (with increasing concentrations to the right). Discrete concentration values of dissolved Mn in
Atexcac and particulate Mn, Fe and S in Alberca, normalized by their respective average were added. Epi-, meta-
and hypo-limnion layers are represented for each lake according to $O_2$ and temperature profiles.


## 4. RESULTS


### 4.1. Lake Alchichica

The water column of Lake Alchichica showed a pronounced stratification compared to previous years at the same
period (Fig. 2, Fig. S2; Lugo et al., 2000; Adame et al., 2008; Macek et al., 2020). The water temperature varied
from about 20 °C at the surface to 15.5 °C at a 30 m depth and below. Conductivity slightly decreased from 13.8
to 13.7 mS/cm between the surface and 20 m in depth (salinity decreasing from 7.9 to 7.8 psu). Dissolved $O_2$ was
saturated at the lake surface (112 % or 7.5 mg/L) and rapidly decreased to 0 mg/L between ~ 10 and 20 m in depth.


The oxidation reduction potential (ORP) signal was stable between 130 and 120 mV from the surface to 30 m and
then decreased down to -270 mV at 60 m of depth. Chlorophyll a averaged 2 µg/L, with a broad peak between ~
7 and 29 m at around 4 µg/L (with a 6 µg/L maximum at 23 m) and then decreased to minimum values (0.5 µg/L)
in the lower water column. Finally, pH remained constant at ~9.2 over the whole water column. Based on these
results, the epi-, meta- and hypolimnion layers of Lake Alchichica in May 2019 extended from 0-10, 10-20 and
20-60 m, respectively (Fig. 2).
Dissolved inorganic carbon (DIC) represented about 95% of all the carbon (DIC+DOC+POC) in the pelagic water
column. Its concentration was almost constant between 34.5 and 35 mM throughout the whole water column except
at 10 m where it significantly decreased to 33 mM (Fig. 3, Table 2). The $\delta^{13}C_{DIC}$ decreased from 2 to ~ 1.5‰
between 5 and 60 m in depth (Fig. 4). The analysis of $DI^{14}C$ content at a depth of 35 m reached 39 % modern
carbon (pMC), equivalent to an apparent age of ~ 7540 years before "present" (i.e. before 1950). Particulate
organic carbon (POC) represented about 0.13 % of the total carbon measured in the water column with a
concentration of 0.07 mM at a 5 m depth, increasing to a maximum of 0.1 mM at 30 m and then decreasing to
~0.02 mM in the bottom part of the water column. The C:N ratio of particulate organic matter (POM) showed a
similar profile with values around 10.5 down to 30 m, progressively decreasing towards 5.9 at 55 m (Fig. 3).
$\delta^{13}C_{POC}$ increased from -26.5 ‰ in the top 30 meters to -24.1 ‰ at 55 m in depth. Dissolved organic carbon (DOC)
represented about 5% of total carbon, with concentrations around 0.5 mM throughout the water column except in
the hypolimnion where it reached up to 5.4 mM. Its isotopic composition varied from -29.3 to -25.1 ‰, with
maximum values found in the hypolimnion (Fig. 4).
The sum and weighted average of total carbon concentrations and isotopic compositions were calculated. The total
carbon concentration depth profile roughly follows that of DOC, while $\delta^{13}C_{total}$ is roughly comprised between 2
and 0 ‰ through the water column, except in the lower part of the hypolimnion where it decreases down to -2.4 ‰
on average (Figs. 3, 4; Table 2).
Total dissolved phosphorus (TDP) is stable down to 20 m where it shows a marked increase from 0.37 to 1.56 µM
at 30 m and then progressively increases up to 3.20 µM at the bottom of the lake (Fig. 5, Table S1). Sulfate
concentration slightly decreases from ~11.8 to 11.7 mM between the surface and 30 m and then increases to
12.2 mM at 60 m. Dissolved Cl followed a similar profile with values around 107 mM at the surface decreasing
below 106 mM at 30 m and increasing back to 111 mM at 60 m (Table S1). Dissolved Mn surprisingly showed
the highest concentrations near the surface (~1.6 µM) before decreasing to ~0.4 µM between 20 and 55 m and
increasing to 1 µM at 60 m. Similarly, dissolved Fe was higher at the lake surface (~0.3 µM) and progressively
decreased near 0 at 50/55m (Fig. 5, Table S1).
In the first 12 cm of sediments, porewater DIC had concentrations (~ 35 mM) and $\delta^{13}C_{DIC}$ (~ 0 ‰) similar to and
slightly lower than the water column values, respectively. Sedimentary organic matter had a $\delta^{13}C_{SOC}$ increasing
from -25.7 to -24.5 ‰ and C:N ratios slightly higher than 10 (Figs. 3, 4; Table S2).
293       **4.2. Lake Atexcac**

Stratification of the Lake Atexcac water column was also very well defined (Fig. 2). Temperature was high (20.6
– 19.6 °C) between 0 and 10 m in depth; it rapidly decreased and remained constant at 16 °C below 20 m.





Conductivity had the same evolution with values around 13 mS/cm near the surface and decreasing to 12.9 mS/cm
under 20 m (salinity decreasing from about 7.44 to 7.3 psu). Dissolved $O_2$ was saturated at the lake surface (115 %,
i.e., 7.6 mg/L) and rapidly decreased to 0 mg/L between ~ 10 and 20 m. The ORP signal was almost constant (~
134 mV) between the surface and 22 m in depth, before decreasing and reaching -175 mV at a 32 m depth.
Chlorophyll a averaged 1 µg/L and showed a narrow peak centered at around 16 m reaching ~2 µg/L, with similar
values at the surface and bottom of the lake (0.8 µg/L). Turbidity showed a pronounced increase below 20 m,
peaking at 23.3 m and returning to surface values at 26 m. Finally, pH remained between 8.80 and 8.85 throughout
the water column. Based on these results, the epi-, meta- and hypolimnion of Atexcac in May 2019 can be broadly
defined as extending from 0-10, 10-20 and 20-39 m, respectively (Fig. 2).
DIC represented about 84 % of all carbon present (DIC+DOC+POC) in the pelagic water column. Its concentration
remained around 26.5 mM from the surface down to 16 m in depth. Below 16 m, DIC decreased in the
hypolimnion, and notably at 23 m where it reached a value of 24.2 mM (Fig. 3, Table 2). The $\delta^{13}C_{DIC}$ was stable
around 0.4 ‰ in the epi-/metalimnion. It markedly increased to 0.9 ‰ at 23 m and reached minimum values
(0.2 ‰) at the bottom of the lake. POC represented about 0.13 % of the total carbon measured in the water column
with concentrations of 0.05 mM in the epi- and metalimnion, decreasing to 0.02 mM in the hypolimnion. The C:N
ratio of POM showed the same depth profile as POC concentration with a value around 9.6 in the epi-/metalimnion
decreasing to 6.6 in the hypolimnion (Fig. 3). $\delta^{13}C_{POC}$ had values around -28.3 ‰ in the epilimnion, showed a
minimum value of -29 ‰ at 16 m and increased to -26.5 ‰ in the hypolimnion. DOC represented about 16% of
total carbon, with a concentration around 1.1 mM throughout the water column except at 16 and 23 m, where it

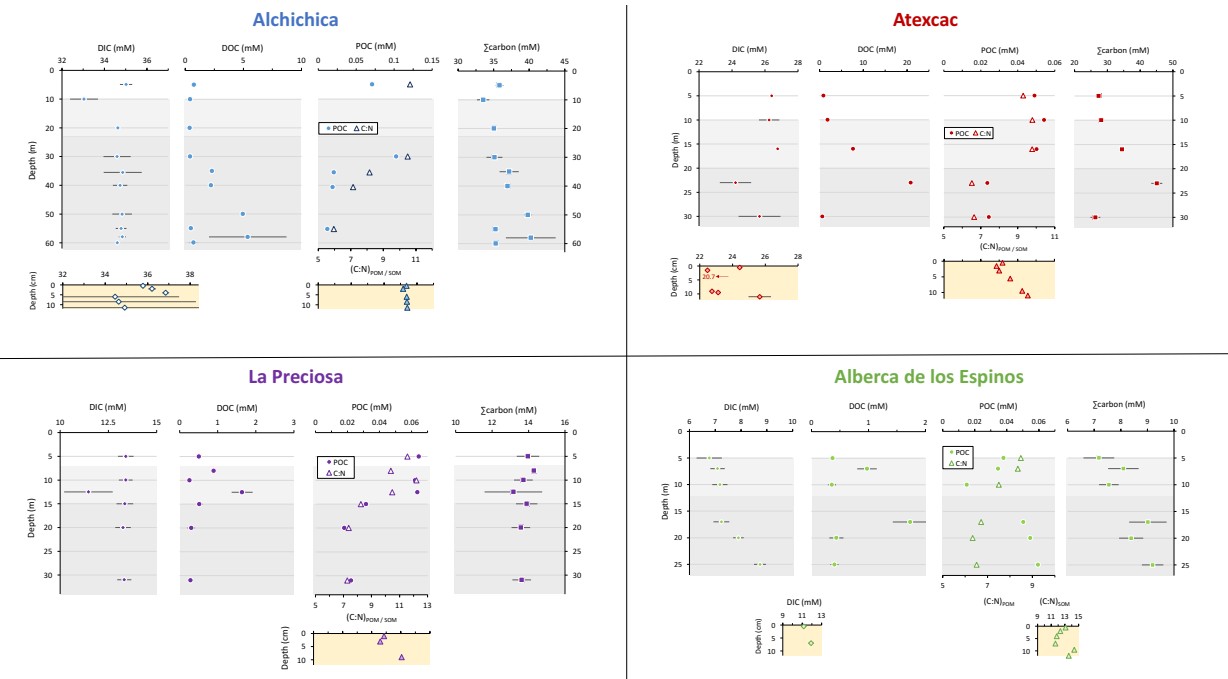

Figure 3. Concentrations in mmoles/L of DIC, DOC, POC and sum of all three reservoirs, C:N molar ratios of POM as a function of depth in the water columns, as well as DIC concentrations in the surficial sediment porewaters and C:N molar ratios of sedimentary OM. Porewaters from La Preciosa's 2016 core were not retrieved.



reached 7.7 and 20.8 mM, respectively. Its isotopic composition showed values increasing from -20 to -9 ‰
between 5 and 23 m, and decreasing to -11 ‰ at 30 m. Total C concentrations and $\delta^{13}C_{total}$ are centered around
27.7 mM and -0.6 ‰ with a clear increase to 38.9 mM and decrease to -2.7 ‰ at 23 m, respectively.
TDP concentrations slightly decreased from ~0.25 µM to 0.19 µM at 16 m, then increased in the hypolimnion to
~ 0.45 µM (Fig. 5; Table S1). Dissolved sulfate concentration was relatively stable at around 2.51 mM throughout
the water column except at 23 m, where it increased to 2.64 mM. Dissolved Cl concentration slightly decreased
from 122 to 121 mM between the surface and 16 m before increasing in the hypolimnion at ~125 mM (Table S1).
Dissolved Mn concentration was constant at 1 µM down to 16 m before dropping to 0 at 23 m and then increasing
again to a maximum value of 2.35 µM at 30 m (Fig. 5; Table S1). This type of profile evolution was also found
for other heavy elements such as Cu, Sr, Ba or Pb among others.
In the first 12 cm of sediments, porewater DIC concentration varied between ~ 21 and 26 mM and $\delta^{13}C_{DIC}$ was
around 0 ‰. Sedimentary organic matter had a $\delta^{13}C_{SOC}$ around -27 ‰ and a C:N ratio increasing from 8 to 10
(Figs. 3, 4; Table S2).

Table 2
Concentrations and isotopic compositions for dissolved inorganic and organic carbon (DIC, DOC), particulate organic carbon (POC) and C:N molar ratios of particulate organic matter (POM). Total carbon concentrations is the sum of all carbon reservoirs measured, $\delta^{13}C_{Total}$ is the weighted average of each $\delta^{13}C$.

| Lake | Sample | DIC | DOC | POC | Total Carbon | (C:N)$_{POM}$ | $\delta^{13}C_{DIC}$ | $\delta^{13}C_{POC}$ | $\delta^{13}C_{DOC}$ | $\delta^{13}C_{Total}$ |
|---|---|---|---|---|---|---|---|---|---|---|
| | | mmoles/L | | | | (molar) | ‰ | | | |
| Alchichica | AL 5m | 35.0 | 0.7 | 0.07 | 35.8 | 10.6 | 2.0 | -26.7 | | 1.4 |
| | AL 10m | 33.0 | 0.4 | | 33.5 | | 2.0 | | -28.3 | 1.6 |
| | AL 20m | 34.6 | 0.4 | | 35.0 | | 1.6 | | -29.3 | 1.3 |
| | AL 30m | 34.6 | 0.4 | 0.10 | 35.1 | 10.5 | 1.7 | -26.3 | -28.3 | 1.2 |
| | AL 35m | 34.9 | 2.3 | 0.02 | 37.2 | 8.1 | 1.6 | -25.7 | -26.8 | -0.2 |
| | AL 40m | 34.7 | 2.2 | 0.02 | 37.0 | 7.1 | 1.6 | -25.1 | -25.8 | -0.1 |
| | AL 50m | 34.8 | 5.0 | | 39.8 | | 1.6 | | -25.1 | -1.8 |
| | AL 55m | 34.8 | 0.5 | 0.01 | 35.3 | 5.9 | 1.5 | -24.1 | -27.6 | 1.1 |
| | AL 58m | 34.8 | 5.4 | | 40.2 | | 1.6 | | -27.7 | -2.3 |
| | AL 60m | 34.6 | 0.7 | | 35.3 | | 1.5 | | -26.1 | 1.0 |
| Atexcac | ATX 5m | 26.4 | 0.92 | 0.05 | 27.4 | 9.3 | 0.4 | -28.4 | -20.0 | -0.4 |
| | ATX 10m | 26.2 | 1.8 | 0.05 | 28.1 | 9.8 | 0.4 | -28.2 | -15.5 | -0.7 |
| | ATX 16m | 26.8 | 7.8 | 0.05 | 34.7 | 9.8 | 0.3 | -29.0 | | 0.2 |
| | ATX 23m | 24.2 | 21.0 | 0.02 | 45.2 | 6.5 | 0.9 | -26.7 | -8.7 | -3.6 |
| | ATX 30m | 25.7 | 0.7 | 0.02 | 26.4 | 6.6 | 0.2 | -26.4 | -11.2 | -0.1 |
| La Preciosa | LP 5m | 13.4 | 0.5 | 0.06 | 14.0 | 11.6 | 0.1 | -26.4 | -27.2 | -0.9 |
| | LP 8m | | 0.9 | 0.07 | | 10.4 | | -27.1 | -20.0 | |
| | LP 10m | 13.4 | 0.3 | 0.06 | 13.7 | 12.2 | 0.2 | -27.4 | -15.5 | -0.4 |
| | LP 12.5m | 11.5 | 1.6 | 0.06 | 13.2 | 10.5 | -0.2 | -27.1 | | -2.8 |
| | LP 15m | 13.4 | 0.5 | 0.03 | 13.9 | 8.2 | -0.3 | -23.5 | -8.7 | -1.3 |
| | LP 20m | 13.3 | 0.3 | 0.02 | 13.6 | 7.4 | -0.4 | -26.3 | -11.2 | -1.0 |
| | LP 31m | 13.3 | 0.3 | 0.02 | 13.6 | 7.3 | -0.4 | -25.2 | -25.4 | -0.9 |
| Alberca de Los Espinos | Albesp 5m | 6.8 | 0.4 | 0.04 | 7.2 | 8.5 | -2.6 | -27.0 | -26.7 | -3.9 |
| | Albesp 7m | 7.1 | 1.0 | 0.03 | 8.1 | 8.3 | -2.3 | -26.2 | -14.7 | -3.9 |
| | Albesp 10m | 7.2 | 0.4 | 0.02 | 7.6 | 7.5 | -4.1 | -28.3 | -25.2 | -5.1 |
| | Albesp 17m | 7.2 | 1.7 | 0.05 | 9.0 | 6.7 | -3.4 | -29.0 | -26.3 | -7.9 |
| | Albesp 20m | 7.9 | 0.4 | 0.05 | 8.4 | 6.3 | -3.3 | -26.5 | -25.1 | -4.5 |
| | Albesp 25m | 8.7 | 0.4 | 0.06 | 9.2 | 6.5 | -2.0 | -25.7 | -27.2 | -3.2 |


### 4.3. Lake La Preciosa

Lake La Preciosa was also stratified (Fig. 2). The temperature varied from about 20 °C at the surface to 16°C at
15 m. Conductivity had a similar evolution decreasing from 2.24 to 2.22 mS/cm between the surface and 15 m
(salinity decreasing from 1.15 to 1.14 psu). Dissolved O$_2$ was saturated at the lake surface (120 %, i.e., 8.4 mg/L)
and rapidly decreased to 0 between ~ 8 and 14 m. The ORP signal was stable between 213 and 225 mV from the


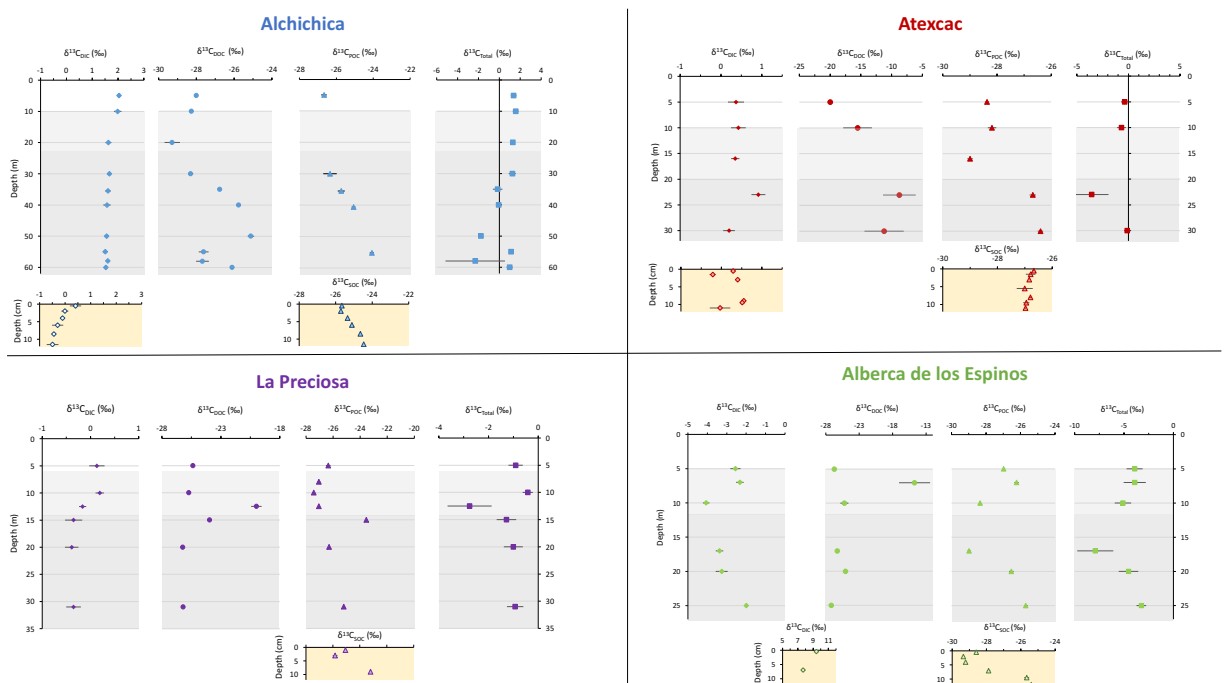

Figure 4. Isotopic compositions of DIC, DOC, POC and weighted average of all three C reservoirs as a function of depth in the water columns as well as isotopic compositions of the porewater-DIC and solid organic carbon from the surficial sediments.

lake surface to 16 m and then decreased down to -105 mV at a 35 m depth. Chlorophyll a concentration averaged
3 µg/L in Lake La Preciosa water column and recorded the highest peak compared to the other lakes, increasing
to about 9 µg/L at 10 m and decreasing below 12 m to reach minimum values (0.7 µg/L) below 15 m. Turbidity
showed a bimodal profile with low values between 0 and 15 m, a large peak between 16 and 19 m and relatively
high values downward. Finally, pH showed a small decrease from 9 to 8.8 between the surface and 15 m. Based
on these results, epi-, meta- and hypolimnion layers of La Preciosa in May 2019 can be broadly defined as
extending from 0-6, 6-15 and 15-45m, respectively (Fig. 2).
DIC represented about 97% of all carbon present (DIC+DOC+POC) in La Preciosa water column. Its concentration
was constant throughout the water column at 13.3 mM, with an exception at 12.5 m, where DIC decreased to
11.5 mM (Fig. 3, Table 2). The $\delta^{13}C_{DIC}$ decreased from about 0.5 ‰ to -0.36 ‰ between the surface and the
hypolimnion. POC represented about 0.3% of the total carbon measured in the water column with a concentration
of 0.06 mM in the epi- and metalimnion, decreasing to 0.02 mM in the hypolimnion. The C:N ratio of POM showed
a very similar depth profile with a value around 11.2 in the epi-/metalimnion decreasing to 7.6 in the hypolimnion.
$\delta^{13}C_{POC}$ decreased from -26.4 to -27.4 ‰ between 5 and 10 m and peaked at -23.5 ‰ at 15 m before returning to
value close to -25 ‰ downward. DOC represented about 3% of the total carbon, with a concentration around
0.5 mM throughout the water column except at 15 m where it peaked at 1.6 mM. $\delta^{13}C_{DOC}$ was mostly around -26
‰ except between 10 and 12.5 m, where it reached up to -20 ‰. The total C concentration was relatively stable
at ~13.8 mM, while $\delta^{13}C_{total}$ was centered around -1 ‰ with a decrease down to -2.8 ‰ at 12.5 m.



TDP was stable at ~ 0.21 μM between 5 and 12.5 m and increased in the hypolimnion to 0.31 μM (Fig. 5, Table
S1). Dissolved sulfate concentration slightly decreased from ~1.22 to 1.15 mM between the surface and 12.5 m
and was stable at ~1.16 mM downward. The total S concentration remained stable throughout the water column at
a value of ~1.19 mM. Dissolved Cl followed a similar profile with a value around 8.4 mM at the surface decreasing
to ~7.8 mM below 12.5 m (Table S1). Dissolved Mn was around 1 μM at 5 m, decreased to 0.3 and 0.6 μM
between 8 and 15 m and increased back to values above 1 μM below that. Dissolved Fe was above detection limit
(~0.1 μM) at a 5 m depth (0.12 μM) only (Fig. 5; Table S1).
In the first 10 cm of the sediments, $\delta^{13}C_{SOC}$ values increased downwards from ~ -25.5 to -23.2 ‰ and C:N ratios
from 9.8 to 11 (Figs. 3, 4; Table S2). Porewaters from the 2016 core were not retrieved.

**4.4. Lake La Alberca de los Espinos**
Stratification of the water column in La Alberca de los Espinos was also well defined (Fig. 2). Temperature was
around 23 °C between 0 and 5 m in depth; it rapidly decreased to 18.2 °C at 12 m and slowly decreased down to
16.5 °C at 26 m. Conductivity was at 1.20 mS/cm down to 6 m, and decreased to 1.17 mS/cm down to 16 m before
increasing to 1.27 mS/cm at 26 m (salinity between 0.58 and 0.64 psu). Dissolved $O_2$ was saturated at the lake
surface (118 %, i.e., 7.9 mg/L) and rapidly decreased to 0 between ~ 5 and 12 m. The ORP signal was mostly
comprised between 160 and 170 mV between the surface and 17 m before decreasing to -65 mV at 21 m and -92
mV at 26 m.  La Alberca had relatively high chlorophyll a levels throughout the water column (3.1 μg/L on
average) but showed at least three distinctive peaks, all reaching approximately 4 μg/L. They were found (i)
between 6 and 9.5 m, (ii) at around 12.5 m and (iii) between 16 and 19 m. The turbidity profile showed a
pronounced increase from 16 to 19 m, and a slight decrease towards the sediment-water interface. The pH showed
relatively important variations from 9.15 at the lake surface to 8.75 between 6.5 and 10 m, further decreasing to
7.5 between 16 and 26 m. Based on these results, epi-, meta- and hypolimnion layers of Lake La Alberca de los
Espinos in May 2019 can be defined as extending from 0-5, 5-12 and 12-30 m, respectively (Fig. 2).
DIC represented about 91 % of all carbon present (DIC+DOC+POC) in the water column. Its concentration
progressively increased from 6.8 mM at 5 m to 7.2 mM between 10 and 17 m. It further increased to 8.7 mM down
to 26 m (Fig. 3, Table 2). The $\delta^{13}C_{DIC}$ was about -2.4 ‰ between the surface and 7 m in depth, decreased to -4.1
‰ at 10 m, and increased back up to -2 ‰ at 25 m. POC represented about 0.5 % of the total carbon measured in
the water column with a concentration of 0.04 mM at the surface decreasing to 0.02 mM at 10 m and increasing
back to 0.05 mM in the hypolimnion. C:N ratio of POM progressively decreased from 8.5 at the surface to below
6.5 in the hypolimnion. $\delta^{13}C_{POC}$ had minimum values at 10 and 17 m (-28.3 and -29 ‰, respectively). Above and
below, $\delta^{13}C_{POC}$ was around -26.4 ‰. DOC represented about 8 % of the total carbon, with a concentration around
0.4 mM throughout the water column except at 7 and 17 m where DOC peaked to 1 and 1.7 mM, respectively (Fig.
3). Its isotopic composition was mostly comprised between -27 and -25 ‰ except at 7 m where it reached -15 ‰
(Fig. 4). Total C concentration increased downward from about 7 to 9 mM (Fig. 3). $\delta^{13}C_{total}$ decreased from -3.9
to -7.9 ‰ between 5 and 17 m and then increased up to -3.2 ‰ at 25 m (Fig. 4).
Total dissolved phosphorus increased from 2.9 to 27.4 μM between 5 and 25 m (Fig. 5, Table S1). Dissolved
sulfates as measured by chromatography were only detectable at 5 m with a low concentration of 12 μM, while



total dissolved S measured by ICP-AES showed values in the hypolimnion higher than in the upper layers (~ 10.3
vs 7.4 μM, Table S1). Dissolved Cl slightly decreased from 4.25 to 4 mM between 5 and 10 m, before increasing
back to 4.2 mM at 25 m. Dissolved Mn concentrations decreased from 1.5 to 0.5 μM between 5 and 10 m, then
increased to 2 μM at 25 m. Aqueous Fe was only detectable at 25 m with a concentration of 0.23 μM (Fig. 5, Table
S1). In parallel, particulate S concentrations increased with depth, with a marked increase from 0.1 to 0.6 μM
between 20 and 25m. This was spatially correlated with a 25-fold increase in particulate Fe (from 0.2 to 5.97 μM).
Particulate Mn showed a peak between 17 and 20 m around 1 μM, contrasting with values lower than 0.06 μM
above 10 m and lower than 0.15 below 20 m (Fig. 2, Table S3).
In the first centimeters of the sediments, porewater DIC concentration and $\delta^{13}C_{DIC}$ varied between ~ 11 and 12 mM
and between +8 and +10 ‰, respectively. Sedimentary organic matter had a $\delta^{13}C_{SOC}$ globally increasing from -
29.4 to -25.5 ‰ and a C:N ratio varying between 11.6 and 14.3 (Figs. 3, 4; Table S2). Surficial sedimentary
carbonates had a $\delta^{13}C_{CaCO3}$ around -1.5 ‰.

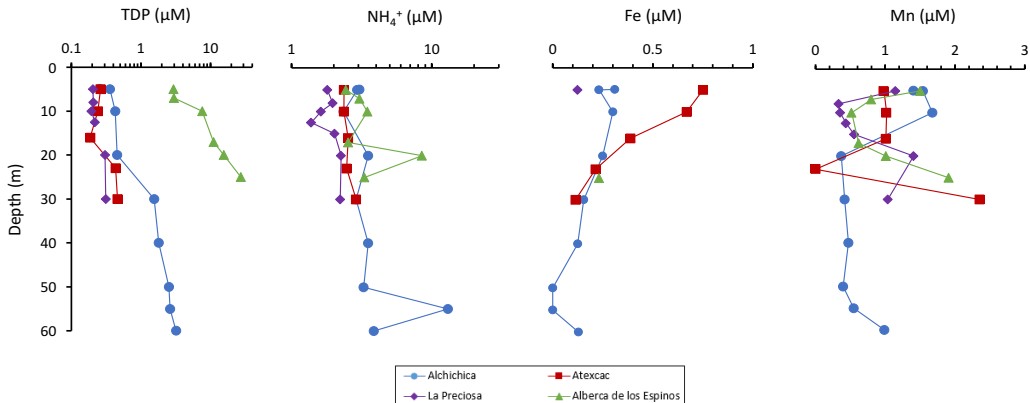

Figure 5. Concentrations of dissolved nutrients in micromoles.L$^{-1}$ in the water columns of the four lakes as a
function of depth. TDP and TDS stands for 'total dissolved P' and S', respectively, and were measured by ICP-
AES. Fe and Mn were measured by ICP-MS. Nitrogen species were measured by colorimetry.


**5.  DISCUSSION**

**5.1. General factors influencing the C cycle across the Mexican crater lakes**

**5.1.1.    The alkalinity gradient**
Salinity and alkalinity (roughly equal to DIC) gradually increase from Lake La Alberca de los Espinos (~0.6 psu
and 7 mM) to Alchichica (~7.9 psu and 35 mM), while lakes La Preciosa and Atexcac have intermediate values of
1.15 and 7.44 psu and 13 and 26 mM, respectively (Table 1 and 2, Zeyen et al., 2021). The DIC gradient along
which these four lakes distribute can be linked with different concentration stages of an initial dilute alkaline water
(Zeyen et al., 2021), those different concentration stages being ultimately controlled by the different hydrological
regimes of the lakes. First, the weathering of Cretaceous limestone in the SOB (with a $\delta^{13}C$ around 2 ± 1 ‰;
Armstrong-Altrin et al., 2011; Núñez Useche et al., 2014) together with basaltic/andesitic bedrock (Armienta et





al., 2008; Carrasco-Núñez et al., 2007; Lelli et al., 2021) favors the inflow of more alkaline and DIC-concentrated
groundwaters than in Lake La Alberca which lies on a purely basaltic basement (Rendon-Lopez, 2008; Siebe et
al., 2014; Zeyen et al., 2021). Second, the SOB area presently experiences higher evaporation than precipitation
rates (Alcocer, 2021), probably playing an important role in concentrating solutes and decreasing the water level
in lakes Atexcac, Alchichica and La Preciosa (Anderson and Stedmon, 2007; Zeyen et al., 2021). Consistently,
substantial "sub-fossil" microbialite deposits emerge above the current water level in lakes Alchichica and
Atexcac, evidencing some significant lake level decrease (by up to 15 m in Lake Atexcac, *i.e.* ~40% of today's
lake maximum depth; and by about 3 m in Alchichica). Patches of emerged microbialites are also found in Lake
La Preciosa. By contrast, emerged microbialites are almost not observed in Lake La Alberca de los Espinos
(Fig. S1).
Additional local parameters, such as varying groundwater paths and fluxes (Furian et al., 2013; Mercedes-Martín
et al., 2019; Milesi et al., 2020; Zeyen et al., 2021), most likely play a role in explaining part of the variations in
DIC concentration between lakes. For example, Lake La Preciosa's water composition significantly differs from
that of Lake Alchichica and Atexcac, despite a similar geological context and climate (all located within 50 km$^2$,
Fig. 1). This could be explained by the fact that groundwaters in the SOB area become more saline as they flow
towards the center of the basin and through the crater lakes (Silva Aguilera, 2019; Alcocer, 2021). Since
groundwaters flow through La Preciosa first, they are more concentrated as they enter Alchichica than when they
enter La Preciosa (Silva Aguilera, 2019; Alcocer, 2021; Lelli et al., 2021). Distinct regimes of volcanic $CO_2$
degassing into these crater lakes may also contribute to variations of the C mass balance between the four lakes.
Last, different remineralization rates of organic carbon could also be a source of heterogeneity between the lakes
DIC content. However, assuming that all POC and DOC ultimately remineralize into DIC, it would only represent
a relatively small portion of the total carbon (16 % in Lake Atexcac, 9 % in Lake La Alberca de los Espinos and
~5 % for lakes Alchichica and La Preciosa). From an isotopic mass balance perspective, the $\delta^{13}C_{DIC}$ of the three
SOB lakes lie very far from $\delta^{13}C_{POC}/\delta^{13}C_{DOC}$ signatures (Fig. 4), whereas Lake La Alberca exhibits more negative
$\delta^{13}C_{DIC}$, slightly closer to OC signatures (Fig. 4). This latter lake also stands out from the others because of the
dense vegetation which surrounds it (Fig. S1). Therefore, La Alberca seems to be the only lake where OC
respiration could be a significant source of inorganic C.
Mean $\delta^{13}C_{DIC}$ values of the lakes broadly correlate with their alkalinity/salinity. This relationship is expected as
evaporation generally increases the $\delta^{13}C_{DIC}$ of residual water, notably because it increases lake $pCO_2$ and primary
productivity which bolsters $CO_2$ degassing and organic C burial, both having low $\delta^{13}C$ compared to DIC (e.g. Li
and Ku, 1997; Talbot, 1990). By controlling the DIC speciation ($H_2CO_3/CO_{2(aq)}$, $HCO_3^-$, $CO_3^{2-}$), pH also strongly
influences $\delta^{13}C_{DIC}$ because there is a fractionation of up to ~9 ‰ between the different DIC species (Emrich et al.,
1970; Mook et al., 1974; Bade et al., 2004). Consistently, the $\delta^{13}C_{DIC}$ of Mexican lakes are in the expected range
for lakes with a pH around 9 (Bade et al., 2004), where DIC is dominated by $HCO_3^-$. However, the pH values of
the studied Mexican lakes are too close to each other to explain the significant difference observed between their
$\delta^{13}C_{DIC}$ (Fig. 4; p=4.2x10$^{-3}$ for Lakes Atexcac and La Preciosa which have the closest $\delta^{13}C_{DIC}$). Last, lakes with
lower DIC concentrations are expected to have a $\delta^{13}C_{DIC}$ more easily influenced by exchanges with other carbon
reservoirs, such as organic carbon (through photosynthesis/respiration) or other DIC sources (e.g., depleted
volcanic $CO_2$ or groundwater DIC) – compared with buffered, high DIC lakes (Li and Ku, 1997; Fig. S3). This
illustrates another (indirect) influence of the inter-lake chemical gradient on $\delta^{13}C_{DIC}$.





Therefore, the alkalinity gradient and to a first order, the size, isotopic composition and responsiveness of the DIC
reservoir to biogeochemical processes are controlled by the local hydro-physico-chemical parameters of the lakes.

### 5.1.2. Stratification of the lakes

Stratified water columns can sustain strong physico-chemical gradients, where a wide range of biogeochemical
reactions impacting the C cycle can take place (e.g. Jézéquel et al., 2016). Here, temperature, conductivity and $O_2$
profiles show that the four lakes were clearly stratified at the time of sampling and had a similar general structure,
although depths defining the successive epi-, meta- and hypolimnion layers differ between the lakes (Fig. 2). For
example, we found a clear offset in Lake Alchichica and Lake La Alberca between the depth of $O_2$ depletion and
the depth below which the ORP decreases. By contrast, ORP sharply dropped below the depth where $O_{2(aq)}$
disappeared in Atexcac and La Preciosa. Meanwhile, in all four lakes the ORP decreased below the depth where
chlorophyll a peaks collapsed. This pigment being a tracer of oxygenic photosynthesis, it suggests that ORP was
buffered at a high value by photosynthetically produced oxygen during C fixation and only decreased at a depth
where aerobic respiration became higher than oxygenic photosynthesis. The offset between the ORP drop and $O_2$
depletion in Lake Alchichica and Alberca could result from more extended peaks of chlorophyll a that we can
observe in these two lakes (Fig. 2). The exact factors causing this distribution of oxygenic primary producers
remain to be determined. In the end nonetheless, this impacts the depth distribution of other microbial metabolisms
that thrive at different redox levels as well as the depths at which authigenic particles precipitate following redox
reactions, as exemplified by the depth profiles of turbidity and the particulate metal concentrations (Fig. 2).
The evolution of pH with depth is another example of the interplay between physico-chemical stratification of the
lakes and their respective C cycle. pH showed a stratified profile in La Preciosa and La Alberca, whereas it
remained constant in Alchichica and Atexcac. The pH decline at the oxycline of Lake La Preciosa was associated
with the decrease of DOC, POC and chlorophyll a concentrations and $\delta^{13}C_{DIC}$ values, reflecting the high impact of
oxygen respiration (i.e. carbon remineralization) at this depth (Figs. 2-4). In Lake La Alberca, the surface waters
are markedly more alkaline than the bottom waters, with a two-step decrease of pH occurring at around 8 m and
17 m (total drop of 1.5 pH unit). Based on the same observations as in La Preciosa, this likely results from high
OM respiration, although input of volcanic acidic gases (e.g. dissolved $CO_2$) might also contribute to the pH
decrease in the bottom waters, as reflected by negative $\delta^{13}C_{DIC}$ signatures and the increase of [DIC] and
conductivity in the hypolimnion (Figs. 2-4). By contrast, while the same pieces of evidence for oxygen respiration
([POC], chlorophyll, etc.) can be detected in the two other lakes, this did not similarly impact their pH profile
(Fig. 2). This suggests that the acidity generated by these reactions is buffered by the much higher alkalinity
measured in these two lakes. Thus, external constraints on the alkalinity buffering capacity of these lakes (e.g.,
lake hydrology, fluid sources, Sect. 5.1.1) influence their vertical pH profile, which is particularly important
considering the critical interplay between pH and biogeochemical reactions affecting the C cycle (e.g. Soetaert et
al., 2007).
In summary, although the four lakes present the same general structure and environmental conditions, external
factors (such as hydrology, fluid sources or stratification characteristics) result in contrasting compositions of their



water chemistries, which in turn, has a critical impact on the physico-chemical depth profiles of each lake and their
biogeochemical carbon cycle functioning.

### 5.2. From water column primary production to sedimentary organic matter: insights from POC and DIC signatures

In this section, we discuss the different biological processes that can be evidenced based on the depth variations
of DIC and POM chemical and isotopic compositions.

#### 5.2.1. Primary productivity by oxygenic photosynthesis in the upper water column

All four crater lakes are endorheic basins, i.e. there is no surface water inflow or outflow. Therefore, the organic
carbon sources are predominantly autochthonous, mainly resulting from planktonic autotrophic C fixation. This is
supported by C:N ratios of POM that were comprised between 6 and 12 in the four lakes, i.e., close to the Redfield
but far from land plant ratios. Abundant vegetation covers the crater walls of Lake Alberca and to a lesser extent
Lake Atexcac; some plant debris were observed and sampled in the sediment cores of these two lakes. They had
high C:N ratios, typical of plant tissues (between 24 and 68) and significantly higher than those of the bulk organic
matter of surficial sediments (between 8 and 13) and the water column (between 6 and 12) (Fig. 3). Therefore,
local allochtonous organic carbon in these two lakes – albeit present – does not significantly contribute to their
bulk organic signal.
The importance of planktonic autotrophic C fixation as a major source of organic C in the four lakes is further
supported by the assessment of the isotopic discrimination between DIC and organic biomass, expressed as
$\Delta^{13}C_{POC-DIC}$ and $\varepsilon_{POC-CO2}$ (Table 3). $\Delta^{13}C_{POC-DIC}$ vary between -29 and -23 ‰ (corresponding to $\varepsilon_{POC-CO2}$ between -
20 and -14 ‰; Table 4) throughout the four water columns, which is in the typical range of planktonic oxygenic
phototrophs (Pardue et al., 1976; Sirevag et al., 1977; Thomas et al., 2019). Yet, these values exhibit variability –
both within a single water column (up to 4.5 ‰) and between the different lakes (up to 6 ‰, Figs 4 and 6) – which
could trace several abiotic and biotic factors.
Notably, higher DIC availability in Alchichica and Atexcac probably makes the carboxylation step more limiting
during photosynthesis (e.g. O'Leary, 1988; Descolas-Gros and Fontungne, 1990; Fry, 1996) and increase $|\Delta^{13}C_{POC-}$
$_{DIC}|$ in these lakes compared to La Preciosa and Alberca (Fig. 6a; between 24 and 27 ‰ for these two lakes *versus*
28 to 29.5 ‰ for Alchichica and Atexcac, at the peak of Chl. a). Indeed, lower $CO_{2(aq)}$ availability and/or higher
reaction rates result in transport-limited rather than carboxylation-limited uptake and thus, smaller C isotope
fractionation between POC and DIC (Pardue et al., 1976; Zohary et al., 1994; Fry, 1996; Close and Henderson,
2020). This is because the isotopic fractionation associated with diffusion is much smaller than with carboxylation
and because a higher proportion of the DIC entering the cells is converted into organic biomass (e.g. Fogel and
Cifuentes, 1993). Consistently, we notice a correlation among the lakes between $a(CO_2)_{(aq)}$ (or [DIC]) and $|\varepsilon_{POC-}$
$_{CO2}|$ at depths where oxygenic photosynthetic peaks (Fig. S4). Furthermore, Lakes La Preciosa and Alberca are
considered more eutrophic than the two other lakes (Lugo et al., 1993; Vilaclara et al., 1993; Callieri et al., 2013)
consistently with higher chlorophyll a content and photosynthetic rates and thus smaller $|\Delta^{13}C_{POC-DIC}|$. Additionally,



higher water temperatures in Alberca de los Espinos (by ~ 3 °C) could partly contribute to smaller $|\Delta^{13}C_{POC\text{-}DIC}|$ in
this lake (Sackett et al., 1965; Pardue et al., 1976; Descolas-Gros and Fontungne, 1990).
Unlike $\delta^{13}C_{DIC}$, organic carbon isotopic signatures do not evolve linearly with the alkalinity/salinity gradient,
suggesting other lake- and microbial-specific controls on these signatures. These include: diffusive or active uptake
mechanisms, specific carbon fixation pathways, the fraction of intracellular inorganic carbon released out of the
cells, cell size and geometry (Werne and Hollander, 2004 and references therein) and remineralization efficiency.
Moreover, an increasing number of isotopic data has evidenced a significant variability of the isotopic fractionation
achieved by different purified RuBisCO enzymes ($\epsilon_{RuBisCO}$, Iñiguez et al., 2020), and even by a single RuBisCO
form (Thomas et al., 2019). Thus, caution should be paid to the interpretation of the origin of small isotopic
variations of the biomass in distinct environmental contexts because RuBisCO alone can be an  important source
of this variability (Thomas et al., 2019).

Table 3
Index for mathematical notations used in the text including C isotopic composition of a reservoir X ($\delta^{13}C_X$),
isotopic discrimination between the two carbon reservoirs X and Y ($\Delta^{13}C_{X\text{-}Y}$). In the main text, we report organic
C isotopic discrimination *versus* both bulk DIC ($\Delta^{13}C_{POC\text{-}DIC}$) – in a way to facilitate studies intercomparison and
because it is the commonly reported raw measured data (Fry, 1996) – and calculated $CO_{2(aq)}$ ($\epsilon_{POC\text{-}CO2}$) in order to
discuss the intrinsic isotopic fractionations associated with the lakes metabolic diversity. All C isotope values and
fractionations are reported relative to the international standard VPDB (Vienna Pee Dee Belemnite).

| Symboles | Mathematical Expression | Signification |
|---|---|---|
| $\delta^{13}C_X$ | $\left(\dfrac{\left(\frac{^{13}C}{^{12}C}\right)_X}{\left(\frac{^{13}C}{^{12}C}\right)_{VPDB}} - 1\right) * 1000$ | Relative difference in $^{13}C:^{12}C$ isotopic ratio between a sample of a given C reservoir and the international standard "Vienna Pee Dee Bee", expressed in permil (‰). $\delta^{13}C_{total}$ represents the weighted average of $\delta^{13}C$ for all DIC, DOC and POC. |
| $\Delta^{13}C_{X\text{-}Y}$ | $= \delta^{13}C_X - \delta^{13}C_Y \approx 1000\,ln\,\alpha_{X\text{-}Y}$ | Apparent isotopic fractionation between two reservoirs 'X' and 'Y'. Difference between their measured C isotope compositions approximating the fractionation $\alpha$ in ‰. |
| $\epsilon_{X\text{-}CO2}$ | $= (\alpha_{X\text{-}CO2} - 1)1000 \approx \delta^{13}C_X - \delta^{13}C_{CO2}$ | Calculated isotopic fractionation between a reservoir 'X' and $CO_{2(aq)}$. $\alpha_{X\text{-}CO2}$ is calculated as $(\delta^{13}C_X+1000)/(\delta^{13}C_{CO2}+1000)$ where $\delta^{13}C_X$ is measured and $\delta^{13}C_{CO2}$ is computed based on DIC isotopic composition and speciation (see supplementary information). |
| Indexes | DIC<br>DOC<br>POC<br>SOC | Dissolved Inorganic- ,<br>Dissolved Organic- ,<br>Particulate Organic- ,<br>Sedimentary Organic-Carbon |



### 5.2.2. Aerobic respiration at the oxycline

At the oxycline of stratified water bodies, aerobic respiration of OM by heterotrophic organisms favors the
transition from oxygenated upper layers to anoxic bottom waters. In the water column of the four lakes, $\Delta^{13}C_{POC\text{-}}$
$_{DIC}$ (and $\epsilon_{POC\text{-}CO2}$) show increasing values in the hypolimnion, and especially below the chlorophyll a peaks (Fig. 2;
6). This trend also correlates with increasing $\delta^{13}C_{POC}$, decreasing $(C:N)_{POM}$ ratios as well as decreasing POC



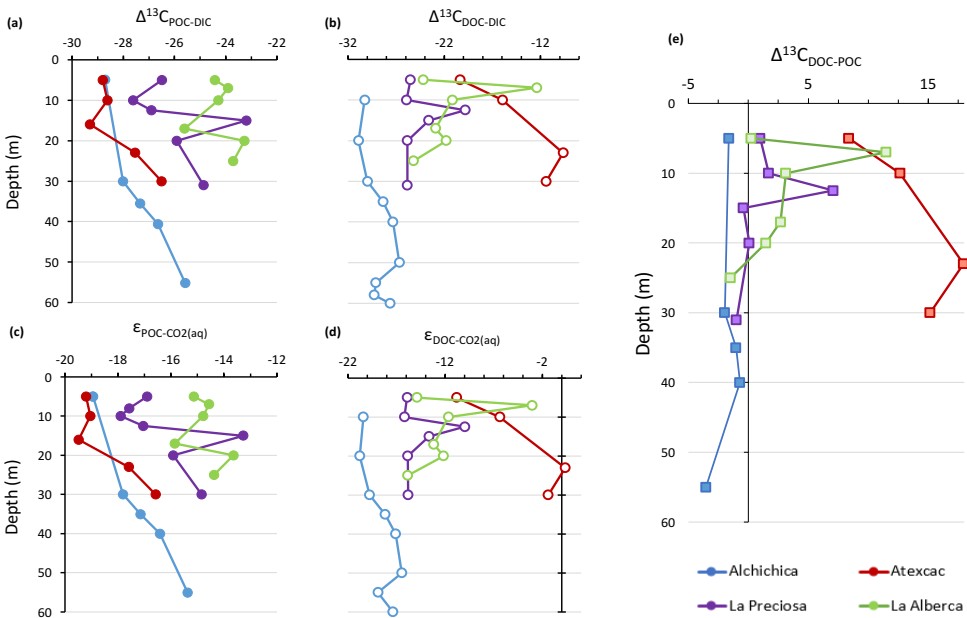

Figure 6. Isotopic fractionations between POC and DIC, DOC and DIC, DOC and POC in the water columns of the four lakes, expressed as $\Delta^{13}C_{x-y}$. Refer to Table 3 for more detail about the $\Delta$ notation.

concentrations (except in La Alberca) (Figs 3 and 4). Decreasing POC concentrations near the oxycline and
redoxcline are consistent with the fact that part of the upper primary production is degraded deeper in the water
columns and/or that there is less primary production in the anoxic bottom waters. Increase of $\delta^{13}C_{POC}$ in the
hypolimnion of the lakes is consistent with heterotrophic activity and points out that POC at these depths could
mainly record secondary production rather than sinking degraded primary production. Indeed, heterotrophic
bacteria preferentially grow on available $^{13}$C-enriched amino acids and sugars, thus becoming more enriched than
their C source (Williams and Gordon, 1970; Hayes et al., 1989; Zohary et al., 1994; Briones et al., 1998; Lehmann
et al., 2002; Jiao et al., 2010; Close and Henderson, 2020). Decreasing C:N ratios in POM also reinforce that
conclusion since secondary heterotrophic bacteria biomass generally have C:N between 4 and 5 (Lehmann et al.,
2002). By contrast, residual OM from primary producers degraded by heterotrophs would carry higher C:N
signatures (van Mooy et al., 2002; Buchan et al., 2014) that are not recorded by POM in the lower water columns
of the lakes (Fig. 3). In Lake La Preciosa, the water column shifts from a highly oxygenated state to anoxia in a
~5 m interval against more than 10 m for Alchichica and Atexcac. This correlates with a sharp $\delta^{13}C_{POC}$ increase (+
3.4‰) at 15 m, highlighting how efficient and O$_2$-dependent the remineralization process is in this lake.
The $\delta^{13}C_{DIC}$ signatures in lakes Alchichica and La Preciosa are consistent with the mineralization of OM as they
exhibit lower values below the oxycline than in surficial waters (Figs 2; 4). Similarly to what is observed in several
other water bodies and notably stratified water columns such as the Black Sea (e.g. Fry et al., 1991), surface
photosynthesis increases $\delta^{13}C_{DIC}$ by fixing light DIC, while respiration transfers light OC back to the DIC pool at
depth. Such a decrease of the $\delta^{13}C_{DIC}$ can also be seen in the oxycline of Lake La Alberca between 7 and 10 m.





Table 4
Isotopic fractionations between dissolved inorganic and organic carbon (DIC, DOC) and particulate organic carbon
(POC), where $\Delta^{13}C_{x-y} = \delta^{13}C_x - \delta^{13}C_y$ is the apparent fractionation and $\varepsilon$ is computed as the actual metabolic isotopic
discrimination between $CO_2$ and POC/DOC (see Table 3). $\delta^{13}C_{DOC}$ was not measured at 5 m depth and its value at
10 m was used in this calculation of $\Delta^{13}C_{DOC-POC}$.

| Lake | Sample | $\Delta^{13}C_{POC-DIC}$ | $\Delta^{13}C_{DOC-DIC}$ | $\Delta^{13}C_{DOC-POC}$ | $\varepsilon_{POC-DIC}$ | $\varepsilon_{DOC-DIC}$ |
|---|---|---|---|---|---|---|
| | | ‰ | | | ‰ | |
| Alchichica | AL 5m | -28.7 | | -1.6* | -19.1 | |
| | AL 10m | | -30.3 | | | -20.4 |
| | AL 20m | | -30.9 | | | -20.6 |
| | AL 30m | -28.0 | -30.0 | -2.0 | | -20.9 |
| | AL 35m | -27.3 | -28.4 | -1.0 | -17.9 | -19.9 |
| | AL 40m | -26.6 | -27.3 | -0.7 | -17.3 | -18.3 |
| | AL 50m | | -26.7 | | -16.5 | -17.2 |
| | AL 55m | -25.6 | -29.1 | -3.5 | | -16.6 |
| | AL 58m | | -29.3 | | -15.5 | -19.0 |
| | AL 60m | | -27.6 | | | -17.5 |
| Atexcac | ATX 5m | -28.8 | -20.4 | 8.4 | -19.3 | -10.9 |
| | ATX 10m | -28.6 | -16.0 | 12.6 | -19.1 | -6.5 |
| | ATX 16m | -29.3 | | | -19.5 | |
| | ATX 23m | -27.5 | -9.7 | 17.9 | -17.6 | 0.3 |
| | ATX 30m | -26.5 | -11.4 | 15.2 | -16.6 | -1.5 |
| La Preciosa | LP 5m | -26.5 | -25.5 | 1.0 | -16.9 | -15.9 |
| | LP 10m | -27.6 | -25.9 | 1.7 | -17.9 | -16.2 |
| | LP 12.5m | -26.9 | -19.8 | 7.1 | -17.1 | -10.0 |
| | LP 15m | -23.2 | -23.6 | -0.4 | -13.3 | -13.7 |
| | LP 20m | -25.9 | -25.8 | 0.1 | -15.9 | -15.9 |
| | LP 31m | -24.9 | -25.8 | -1.0 | -14.9 | -15.8 |
| La Alberca de Los Espinos | Albesp 5m | -24.4 | -24.2 | 0.2 | -15.2 | -15.0 |
| | Albesp 7m | -23.9 | -12.4 | 11.5 | -14.6 | -3.1 |
| | Albesp 10m | -24.3 | -21.2 | 3.1 | -14.8 | -11.7 |
| | Albesp 17m | -25.6 | -22.9 | 2.7 | -15.9 | -13.2 |
| | Albesp 20m | -23.3 | -21.8 | 1.5 | -13.6 | -12.2 |
| | Albesp 25m | -23.7 | -25.2 | -1.5 | -14.4 | -15.9 |


### 5.2.3. Primary production in the hypolimnion

Anoxygenic autotrophs commonly thrive in anoxic bottom waters of stratified water bodies (e.g. (Pimenov et al.,
2008; Zyakun et al., 2009; Posth et al., 2017; Fulton et al., 2018; Havig et al., 2018). They have been identified at
different depths in the four Mexican lakes (Macek et al., 2020; Iniesto et al., in press). Based on our results obtained
on samples collected during the stratification period, anoxygenic autotrophs appear to have an impact on the C
cycle of lakes Atexcac and La Alberca only. Lake Atexcac records a concomitant decrease of [DIC] and increase
of $\delta^{13}C_{DIC}$ in the anoxic hypolimnion at 23 m, below the peak of chlorophyll a, suggesting autotrophic C fixation
by chemoautotrophy or anoxygenic photosynthesis. The calculated $\varepsilon_{POC-CO2}$ at 23 m (-17.5 ‰) is consistent with
C isotopes fractionation by purple- and green sulphur-anoxygenic bacteria (PSB and GSB), while $\varepsilon_{POC-CO2}$ in La
Alberca's hypolimnion (~ -15 ‰) is closer to GSB canonical signatures (Posth et al., 2017 and references therein)
(Fig. 6c). In La Alberca, anoxygenic primary productivity is moreover suggested by increasing POC



concentrations. Besides, we also observe a Chl. a peak in the anoxic hypolimnion of this lake (Fig. 2), which likely
represents a bias of the probe towards some bacteriochlorophylls typical of GSB (see supplementary information).
We notice that in Lake Atexcac, C fixation at 23 m by anoxygenic autotrophs causes a shift in the DIC reservoir,
while oxygenic photosynthesis at 16 m does not, suggesting that anaerobic autotrophs are the main autotrophic
metabolisms in this lake (in terms of DIC uptake). In La Alberca, the increase of [POC] to maximum values at
depth also supports the predominance of anoxygenic versus oxygenic autotrophy (Fig. 3). This is similar with other
stratified water bodies which exhibit primary production clearly dominated by anoxygenic metabolisms (Fulton et
al., 2018).
Furthermore, at 23 m in Lake Atexcac and 17 m in Lake La Alberca, we find a striking turbidity peak precisely
where the redox potential and concentrations of dissolved Mn drop (Fig. 2). In Lake Atexcac concentrations of
dissolved metal such as Cu, Pb or Co also drop at 23 m (Fig. S5). In La Alberca, a peak of particulate Mn
concentrations is also detected at 15 m (Fig. 2; data not available for Atexcac). This is most likely explained by
the precipitation of Mn as oxide particles where reduced bottom waters meet oxidative conditions prevailing in
the upper waters. Such Mn-oxides, even at low μM concentrations, can catalyze abiotic oxidation of sulfide to
sulfur compounds (van Vliet et al., 2021), which in turn can be used and further oxidized to sulfate by phototrophic
or chemoautotrophic sulfur-oxidizing bacteria. This is also consistent with the small in increase of $[SO_4^{2-}]$ observed
at 23 m in Atexcac (Table S1). Besides Mn-oxides can be used as electron acceptors during chemoautotrophy
(Havig et al., 2015; Knossow et al., 2015; Henkel et al., 2019; van Vliet et al., 2021).

**5.2.4.  Influence of methanogenesis and volcanic-CO$_2$ degassing from the sediments of Lake La**
**Alberca de los Espinos**
Lake La Alberca shows the least saline/alkaline water column and most peculiar geochemical depth profiles among
the four lakes. Notably, its [DIC] and $\delta^{13}C_{DIC}$ (the lowest of the studied lakes) increase from the lower metalimnion
to the hypolimnion and further into the pore waters of the first cm of sediments with $\delta^{13}C_{DIC}$ reaching up to ~11 ‰
(Figs. 3, 4). Consistently, the calculated CO$_2$ partial pressure (P$_{CO_2}$) increases downward from slightly less than 1x
that of atmospheric P$_{CO_2}$ near the lake surface up to almost 40x at the bottom of the lake (Table S4). The total
carbon concentration depicts a clear increase from surface waters to the bottom of the lake (Fig. 3).
While the increase of [POC] at depth may contribute to the observed $\delta^{13}C_{DIC}$ increase by mass balance, it should
also lower the [DIC] instead of increasing it. Similarly, the sinking of OC particles at depth followed by their
remineralization into DIC cannot explain those observations since this would lower the $\delta^{13}C_{DIC}$ in the hypolimnion
(Fig. 4). Overall, these observations require that a significant source of inorganic $^{13}C$-rich carbon fuels the bottom
waters of Alberca de los Espinos. The source of heavy carbon most likely results from methanogenesis, which
consumes organic carbon in the sediments and produces $^{13}C$-depleted methane and $^{13}C$-rich carbon dioxide
diffusing upward in the water column (acetoclastic methanogenesis, dominant in lacustrine contexts, Whiticar et
al., 1986). Methanogenesis, as an "alternative" OM remineralization pathway could be favored in Lake La Alberca,
because this lake is relatively rich in OM (notably with high [DOC]), and depleted in SO$_4$ compared with the three
other Mexican lakes (Wittkop et al., 2014; Birgel et al., 2015; Cadeau et al., 2020). Based on the isotopic
compositions of sedimentary organic carbon and porewater DIC in Lake La Alberca, we can tentatively calculate





the methane isotopic signature (see supplementary information). The calculated $\delta^{13}C_{CH4}$ in the first 10 cm of
sediments is between -59 and -56.8 ‰, which is consistent with biogenic methane (Whiticar et al., 1986).
Upward diffusing methane may be either (i) partly lost from the lake's surface (i.e. escaping the system) by
degassing or (ii) totally kept in the water column by complete oxidation (either abiotically by oxygenated surface
waters or biologically by methanotrophic organisms). The oxidation of $CH_4$ in the water column should lead to the
formation of $^{13}C$-depleted carbon dioxide that would mix back with the lake DIC (and notably with heavy
methanogenic $CO_2$ produced at depth) as well as $^{13}C$-depleted biomass (as POC or SOC) if it occurs by
methanotrophy. Thus, the net effect of combined methanogenesis and methane oxidation is expected to (i) generate
a $\delta^{13}C_{DIC}$ gradient from high to low values between the sediment porewaters and the chemocline as proposed
elsewhere (Assayag et al., 2008; Wittkop et al., 2014) and (ii) progressively lower sedimentary $\delta^{13}C_{SOC}$ in case of
methanotrophy. Abiotic oxidation of methane by dioxygen is consistent with the observations that $\delta^{13}C_{DIC}$
decreases from porewaters (~ +10 ‰) to the chemocline (-4 ‰), reaching minimum values where dissolved-$O_2$
starts to appear (Fig. 2). On the other hand, microbial anaerobic methane oxidation (AMO) could occur at 17 m
depth through Mn-oxides reduction (Cai et al., 2021; Cheng et al., 2021) and possibly bacterial sulfate-reduction
closer to the water-sediment interface as inferred for the surficial sediments of meromictic Lake Cadagno (Posth
et al., 2017). Indeed, we observe a net increase of particulate Fe and S concentrations at a depth of 25 m and a
peak of solid sulfide minerals in the surficial sediments (Fig. S6). However, $\delta^{13}C_{SOC}$ and $\delta^{13}C_{POC}$ are far from
calculated $\delta^{13}C_{CH4}$, suggesting that AMO is not a major process in the bottom lake waters and surface sediments
(Lehmann et al., 2004) and thus that methanotrophy is not the main $CH_4$ oxidation pathway in Lake La Alberca.
Alternatively, if some portion of the methane escaped oxidation and degassed out of the lake, $\delta^{13}C_{DIC}$ would likely
be driven to extreme positive values with time, as observed elsewhere (Gu et al., 2004; Hassan, 2014; Birgel et al.,
2015; Cadeau et al., 2020). This is not consistent with the observation that the average $\delta^{13}C_{DIC}$ in Lake La Alberca
is about -3 ‰ (Fig. 4), unless an additional counterbalancing source of DIC to this lake exist. In fact, we notice
that $\delta^{13}C_{total}$ averages -4.8‰ in Lake La Alberca, which is very similar to mantle-$CO_2$ signatures (Javoy et al.,
1986; Mason et al., 2017). A contribution from mantle $CO_2$ degassing in this lake may sustain a high $P_{CO2}$ and
[DIC] at depth and maintain the lakes average $\delta^{13}C_{total}$ close to a mantle isotopic signature and notably away from
extreme positive values (if $CH_4$-escape dominated). Moreover, Lake La Alberca is located on top of a likely active
normal fault (Siebe et al., 2012), which is favorable to the ascent of volcanic gases. It is also possible that volcanic
$CO_2$ degassing is coupled to methanogenesis by $CO_2$ reduction in addition to the acetoclastic one described above.
We observe a strong pH decline at depth in this lake (mostly below 17 m, Fig. 2) which could be fostered by both
the acidic volcanic gases (Pecoraino et al., 2015) and methanogenesis, although other redox and microbial
reactions could impact the pH as well (Soetaert et al., 2007).
Overall, volcanic $CO_2$ could be an important source in the C mass balance of Lake La Alberca. We note however,
that volcanic $CO_2$ alone cannot explain the very positive $\delta^{13}C_{DIC}$ in the sediment porewaters. Only a future
quantification of the fluxes of sedimentary methane production, volcanic $CO_2$ and possible $CH_4$ efflux out of the
lake will help to better constrain the peculiar C cycle of Lake La Alberca.

**5.2.5.   Which OM from the water column transfers to the surficial sediments?**



Although the nature and geochemical signatures of the OM that deposits in the bottom sediments may vary
throughout the year, it is interesting to infer from what part(s) of the water column surficial sedimentary OM comes
from during the stratified seasons.
In the three lakes from the SOB, $\delta^{13}C$ and C:N signatures of the surficial sediments OM lie in between POM
signatures from the upper water columns and from the hypolimnion (Figs. 3, 4). Nonetheless, in Alchichica, top
$\delta^{13}C_{SOC}$ and $(C:N)_{SOM}$ signatures (-25.7 ‰ and 10.5, respectively) lie much closer to values recorded in the upper
water column (~ -26.5 ‰ and 10.5, respectively) implying that the upper oxygenic photosynthesis production is
primarily recorded. In Lake Atexcac on the contrary, $\delta^{13}C_{SOC}$ and $(C:N)_{SOM}$ signatures (~ -27 ‰ and 8,
respectively) lie closer to values recorded in the hypolimnion (~ -26.5 ‰ and 6.5, respectively) suggesting that
SOM records mostly the anaerobic primary production. Finally, in Lake La Alberca, surficial $\delta^{13}C_{SOC}$ are markedly
more negative (by ~ 2 to 3 ‰) than the deepest and shallowest water column values (Fig. 4) but they are close to
what is recorded at the redoxcline depth of 17 m. However, the $(C:N)_{SOM}$ values are much higher than what is
measured in the water column, which is consistent with remineralization of OM by sulfate-reduction and
methanogenesis in sediments of this lake (see also Sect. 1). Therefore, OM biogeochemical signatures in La
Alberca's surficial sediments could mainly reflect the effect of early diagenesis occurring at the water-sediment
interface. Importantly though, methanogenesis/methanotrophy are recorded in the surficial sediments porewaters
(notably seen through extremely positive $\delta^{13}C_{DIC}$) but not in the solid sediments that show neither very negative
$\delta^{13}C_{SOC}$ nor positive $\delta^{13}C_{carbonates}$ in the first 10 cm.
Overall, this suggests that OM depositing at the bottom of these stratified lakes do not always record geochemical
signatures from the same sections of the water columns. Notably, they do not necessarily record the signatures of
primary production by oxygenic photosynthesis from the upper column. For example, in Lake Atexcac,
sedimentary OM records instead primary production by anoxygenic photosynthesis, even though POC
concentration was maximum in the upper water column. In Lake La Alberca, OM is rapidly altered by diagenesis
processes, but the signal of methanogenesis is not preserved in the sedimentary OM or carbonates, but only
recorded by the sediment porewaters.

**5.3. A particularly large and central DOC reservoir**

In all four Mexican lakes studied here, the DOC reservoir occupies a predominant role, while showing quite diverse
dynamics and characteristics between the lakes. Indeed, the four lakes have a high DOC content but very different
[DOC] / $\delta^{13}C_{DOC}$ profiles and signatures despite quite similar ones for the DIC and POC reservoirs (Fig. 3; 4).
Evaporation may be one process increasing DOC concentrations (Anderson and Stedmon, 2007; Zeyen et al.,
2021). However, it is likely marginal here because on the contrary to what was observed for DIC, there is no
correlation between the average [DOC] in the Mexican lakes and their salinity. Moreover, evaporation would not
explain the significant intra-lake DOC depth variability.



In this section, we explore the different patterns of DOC production and fate, which depend on slight environmental
and biological variations between the Mexican lakes. Moreover, we further describe the role of the DOC reservoir
on other processes of the lakes C cycle and its potential implications in past oceans C cycle perturbations.

**5.3.1. Sources and fate of DOC**
Dissolved organic carbon is an operationally defined fraction of aqueous organic carbon (here separated from
particulate organic carbon by filtration at 0.22 μm) within a continuum of organic molecules spanning a large
range of sizes, compositions, degrees of reactivity and bioavailability (Kaplan et al., 2008; Hansell, 2013; Beaupré,
2015; Carlson and Hansell, 2015; Brailsford, 2019). The endorheic nature of the studied lakes allows to specifically
focus on the effects of autochthonous primary production, and notably its effects on the DOC reservoir.
Autochthonous DOC can form through multiple processes broadly including: higher-rank OM degradation
processes such as sloppy feeding by predators, UV photolysis or bacterial and viral cell lysis (Lampert, 1978;
Hessen, 1992; Bade et al., 2007; Thornton, 2014; Brailsford, 2019) as well as passive (leakage) or active
(exudation) release by healthy cells (e.g. Baines and Pace, 1991; Hessen and Anderson, 2008; Thornton, 2014;
Ivanovsky et al., 2020). In general, this C release (either "active" or "passive") tends to be enhanced in nutrient-
limited conditions because some recently fixed C is in excess compared with other essential nutrients such as N or
P (Hessen and Anderson, 2008; Morana et al., 2014; Ivanovsky et al., 2020). Moreover, oligotrophic conditions
tend to limit heterotrophic bacterial activity and thus preserve the DOC stocks (Thornton, 2014; Dittmar, 2015).
In the studied lakes, this may partly explain the trend of increasing DOC concentrations from the more eutrophic
Lake Alberca and La Preciosa's waters (0.7 mM on average) to the more oligotrophic Alchichica (1.8 mM) and
Atexcac's (6.5 mM).

*DOC release by autotrophs*
In the four Mexican lakes, [DOC] depth profiles exhibit one or several peaks standing out from low background
values and occurring both in oxic and anoxic waters (Fig. 3). In La Alberca and La Preciosa they correlate with
chlorophyll a peaks. In the two other lakes, they do not match chlorophyll increase. However, in Atexcac, a
remarkable DOC peak (over 10-fold increase, Fig. 3) occurs at the same depth as anoxygenic photosynthesis (Sect.
5.2.3). These co-occurrences support that a large portion of DOC in these three lakes (at least at these depths) arise
from the release of photosynthetic C fixed in excess. Phytoplankton release of DOM is generally thought to be
carried out by (i) an active "overflow mechanism" (DOM exudation) or (ii) a passive diffusion throughout the cell
membranes. In the first case, DOM is actively released out of the cells as a result of C fixation rates higher than
growth and molecular synthesis rates (e.g. Baines and Pace, 1991). Hence, DOM exudation depends on
environmental factors such as irradiance and nutrient availability (e.g. Morana et al., 2014). Besides, it may serve
"fitness-promoting purposes" such as storage, defense or mutualistic goals (Hessen and Anderson, 2008). In the
second case (passive diffusion), DOC release depends on cells permeability and the outward DOC gradient, and
is more directly connected to the amount of phytoplankton biomass (e.g. Marañón et al., 2004). Thus, any new
photosynthate production insures a steady DOM release rate, regardless of the environmental conditions (Marañón
et al., 2004; Morana et al., 2014). In the studied lakes, the fact that lakes La Preciosa and Alberca have lower DOC





but overall higher chlorophyll a concentrations than Atexcac and Alchichica suggests that DOC production does
not directly relate with phytoplankton biomass and is not passively released. Alternatively, an active DOC release
is bolstered by DOC isotopic signatures (see below). Furthermore, the studied Mexican lakes precisely correspond
to environmental contexts (high irradiance and oligotrophic freshwater bodies) where DOM exudation has been
observed and is predicted (e.g. Baines and Pace, 1991; Morana et al., 2014; Thornton, 2014).
At depths where oxygenic photosynthesis occurs, the DOC over total OC ratio averages approximately 85, 99, 94
and 95 % for lakes Alchichica, Atexcac, La Preciosa and La Alberca, respectively. Release of DOC by primary
producers can be characterized by the percentage of extracellular release (PER), which corresponds to the fraction
of DOM over total (dissolved and particulate) OM primary production (e.g. Thornton et al., 2014). PER is highly
variable and averages about 13% of C biomass over a wide range of environments (e.g. Baines and Pace, 1991;
Thornton, 2014). But values as high as 99% have been reported (see Bertilsson and Jones, 2003). Thus, although
some of the DOC measured in the Mexican lakes may correspond to an older long-term DOC reservoir, these DOC
fractions are consistent with extremely high phytoplankton release rates.
An interesting feature is that DOC peaks associated with primary production (mainly photosynthesis) are
characterized by very positive $\Delta^{13}C_{DOC-POC}$ (from +3 to +18 ‰, Fig. 6b). It should be noticed that a switch from
$CO_{2(aq)}$ to $HCO_3^-$ as an inorganic C source (and their 10 ‰ isotopic difference, e.g. Mook et al., 1974) could not
explain alone the isotopic difference between POC and DOC. The isotopic enrichment of DOC molecules
compared to POC could have different origins. First, it supports that DOC may correspond to new photosynthate
release rather than a product of cell lysis or zooplankton sloppy feeding, since the latter would likely produce
$\delta^{13}C_{DOC}$ close to $\delta^{13}C_{POC}$ values. Second, this heavy DOC could originate from photosynthetic organisms using a
different C-fixation pathway inducing smaller isotopic fractionation. In lakes Atexcac and La Alberca anoxygenic
phototrophic bacteria, and notably GSB, could release important amounts of DOC, especially under nutrient-
limiting conditions (Ivanovsky et al., 2020). In contrary to PSB (another group of anoxygenic phototrophs) or
cyanobacteria which use the CCB pathway, GSB use the reductive citric acid cycle or reverse tricarboxylic-TCA
cycle, which tends to induce smaller isotopic fractionations (between ~ 3-13 ‰, Hayes, 2001). If the DOC
reservoirs in lakes Atexcac and La Alberca's hypolimnion originate from GSB fixed C, then their isotopic
composition ($\varepsilon_{DOC-CO2} \approx -5 \pm 5$ and $\varepsilon_{DOC-CO2} \approx -13$ ‰, respectively) are in good agreement with fractionations found
for this type of organisms in laboratory cultures and other stratified water bodies (Posth et al., 2017). The DOC
and POC signatures would deviate from each other if GSB only marginally participated to the POC reservoir but
released most of the DOC. Third, phytoplankton blooms could specifically release isotopically heavy organic
molecules. For example, carbohydrates could be preferentially released under nutrient-limiting conditions as they
are devoid of N and P (Bertilsson and Jones, 2003; Wetz and Wheeler, 2007; Thornton, 2014). Carbohydrates
typically have [13]C-enriched (heavy) isotopic composition (Blair et al., 1985; Jiao et al., 2010; Close and Henderson,
2020). Yet, this molecular hypothesis would hardly explain the full range of $\Delta^{13}C_{DOC-POC}$ variations measured in
Atexcac and La Alberca according to isotopic mass balance of cell specific organic compounds (Hayes, 2001). At
last, such enrichments require otherwise that DOC and DIC first accumulate in the cells. Indeed, if DOC molecules
were released as soon as they were produced, their isotopic composition should approach that of the biomass (i.e.
$\delta^{13}C_{POC}$, within the range of molecules-specific isotopic compositions), which is not the case. If DIC could freely
exchange between inner and outer cell media, maximum "carboxylation-limited" fractionation (mostly between ~





18 and 30 ‰ depending on RuBisCO form, Thomas et al., 2019) would be **expressed** in all synthetized organic
molecules as represented in Fig. 7a (e.g. O'Leary, 1988; Descolas-Gros and Fontungne, 1990; Fry, 1996), which
is also not what DOC records (see $\varepsilon_{DOC\text{-}CO2}$ in Fig. 6d).
Under the environmental conditions of the studied lakes, i.e., low $CO_2$ quantities relative to $HCO_3^-$, local planktonic
competition for $CO_2$ and low nutrient availability, the activation of intracellular DIC concentrating mechanism
(DIC-CM) is expected (Beardall et al., 1982; Burns and Beardall, 1987; Fogel and Cifuentes, 1993; Badger et al.,
1998; Iñiguez et al., 2020). This mechanism is particularly relevant in oligotrophic aqueous media (Beardall et al.,
1982), where $CO_2$ diffusion is slower than in the air (O'Leary, 1988; Fogel and Cifuentes, 1993; Iñiguez et al.,
2020). DIC-CM have been proposed to reduce the efflux of DIC from the cells back to the extracellular solution.
This internal DIC is eventually converted into organic biomass, thereby drawing the cells isotopic composition
closer to that of $\delta^{13}C_{DIC}$ (Fig. 7; Beardall et al., 1982; Fogel and Cifuentes, 1993; Werne and Hollander, 2004).
However, we suggest that the activation of a DIC-CM could preserve a large $\Delta^{13}C_{POC\text{-}DIC}$ while generating an
apparent fractionation between the DOC and POC molecules instead. Indeed, initially fixed OC would be
discriminated against the heavy C isotopes and incorporated into the cellular biomass (Fig. 7c, '$t_i$'). Further,
following the overflow mechanism scenario, high photosynthetic rates (due to high irradiance, temperature and
high DIC despite low $CO_2$) coupled with low population growth rates and organic molecules synthesis (due to
limited abundances of P, N, Fe, etc.) would result in the exudation of excess organic molecules with heavy $\delta^{13}C_{DOC}$
as they are synthetized from residual internal DIC, which progressively becomes $^{13}C$-enriched (Fig. 7c, '$t_{ii}$'). This
suggests that oligotrophic conditions could be a determinant factor in the generation of significantly heavy
$\delta^{13}C_{DOC}$., and even more if they are coupled to high irradiance.
*DOM accumulation in Lake Alchichica*
From the previous discussion, it appears that environmental conditions of the Mexican lakes might favor an
important phytoplanktonic production of DOM. Alcocer et al. (2014) also proposed that an early spring
cyanobacterial bloom in Lake Alchichica favored the production of DOC in the epilimnion. However, at the time
of sampling, the DOC reservoir in this lake was not correlated with any sizeable autotrophic activity at any depth.
Indeed, the large epilimnetic chlorophyll a peak did not correlate with any changes of [DOC] nor $\delta^{13}C_{DOC}$ (Figs.
2-4). Compared with the other lakes, the geochemical conditions at which chlorophyll a is produced in Alchichica
could have been incompatible with the activation of a DIC-CM and significant DOM exudation. For example,
Alchichica had similar $[CO_{2(aq)}]$ as La Preciosa, but higher P and $NH_4^+$ concentrations (Table S1, S3); Lake La
Alberca had higher P concentrations, but presented similar $[NH_4^+]$ and lower $[CO_{2(aq)}]$. We measure a large DOC
increase in the middle of the anoxic hypolimnion of Lake Alchichica, but it did not correspond to any change in
the DIC reservoir as observed for lakes La Preciosa (at 12.5 m) or Atexcac (at 23 m). Moreover at these depths,
photosynthetic active radiation (PAR) is below 0.1% in Alchichica during the stratified season (Macek et al.,
2020), which might not be sufficient to trigger important anoxygenic phytoplankton DOC release.
The DOC reservoir in Alchichica is characterized by a $\delta^{13}C_{DOC}$ (and $\Delta^{13}C_{DOC\text{-}DIC}$) lower than in the other lakes and
systematically showing $^{13}C$-depleted signatures relative to POC (i.e. $\delta^{13}C_{DOC} < \delta^{13}C_{POC}$; Fig. 6). Thus, if the DOC
increase in Alchichica's hypolimnion resulted from the release of photosynthetic OC like in the other lakes, it was


not associated to the same C isotopes fractionation (e.g. if anoxygenic phototrophs did not actively take up DIC,
Fig. 7a).

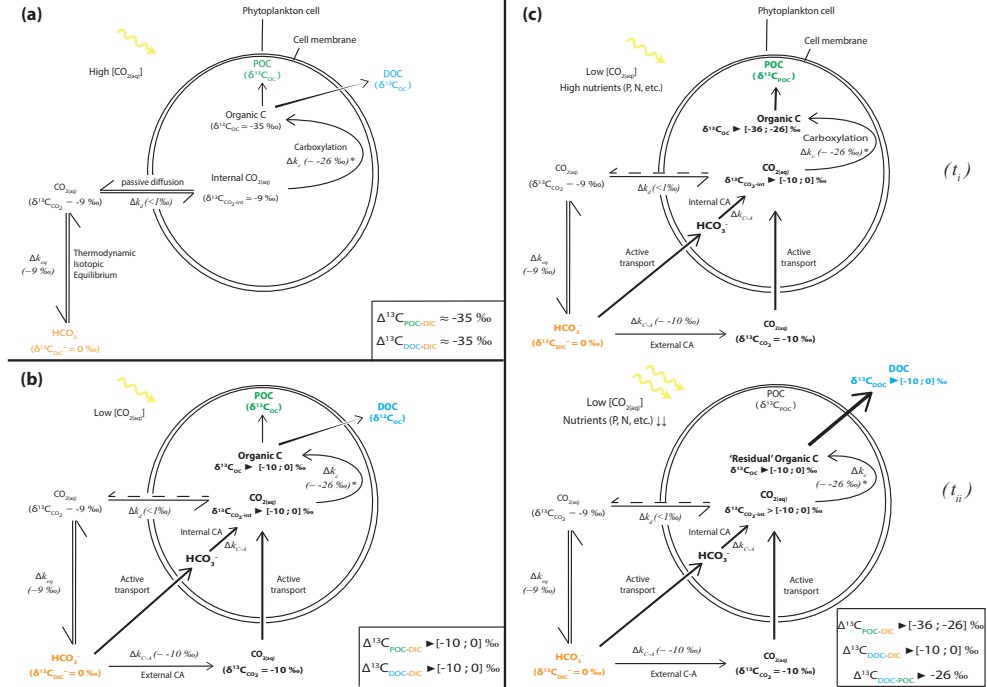



Figure 7. Schematic view of phytoplankton cells during autotrophic C fixation through different C supply
strategies and associated apparent isotopic fractionation between DIC and POC/DOC and between DOC and POC
(*cf*. Table 3). (a) Case where [$CO_{2(aq)}$] is high enough to allow for a DIC supply by passive $CO_{2(aq)}$ diffusion through
the cell membrane and $CO_{2(aq)}$ is at equilibrium with other DIC species. There, isotopic fractionation is maximum
(minimum $\delta^{13}C_{OC}$) because C fixation is limited by the carboxylation step. DOC is released following an in- to
outward cell concentration gradient and has a similar composition to POC. (b) "Classic" view of C isotopic cycling
resulting from active DIC transport within the cell because of low ambient [$CO_{2(aq)}$] (through a DIC-CM). Carbonic
anhydrase (CA) catalyzes the conversion between $HCO_3$ and $CO_{2(aq)}$ inside or outside the cell with an isotopic
fractionation close to equilibrium fractionation (~ 10 ‰). While inward passive $CO_{2(aq)}$ diffusion can still occur,
the DIC-CM activation reduces the reverse diffusion, resulting in internal $CO_{2(aq)}$ isotopic composition
approaching that of the incoming DIC (depending on the fraction of internal $CO_{2(aq)}$ leaving the cell). Acting as a
"closed-system", most of internal DIC is fixed as OC and minimum isotopic fractionation is expressed for both
POC and DOC. (c) Proposed model for C isotopic fractionation with active DIC transport including an isotopic
discrimination between POC and DOC. ($t_i$) Initially fixed C is isotopically depleted and incorporates the cell's
biomass as long as there are sufficient nutrients to enable "complex" organic molecules synthesis. ($t_{ii}$) In low
nutrient conditions, but high photosynthetic activity – fixed OC is released out of the cell as DOC following the
"overflow" hypothesis and inherits heavier isotopic compositions from the residual internal DIC. This leads to
distinct POC and DOC isotopic signatures, with small fractionation between DOC and DIC, the amplitude of which
will depend notably on the rate of $CO_2$ backward diffusion and ratio of biomass C (POC) and released C (DOC).



Alternatively, this hypolimnetic DOC increase could reflect the preservation and accumulation of DOM over the
years in Lake Alchichica, consistently with higher [DOC] measured in 2019 than in the previous years (Alcocer
et al., 2014). While alteration of the DOM reservoir by UV-photolysis would induce a positive isotopic
fractionation (Chomicki, 2009), the slightly negative $\Delta^{13}C_{DOC-POC}$ signatures give support to DOC being mainly a
recalcitrant residual product of primary OM degradation by heterotrophic organisms (Alcocer et al., 2014). Indeed,
the preferential consumption of labile $^{13}$C-enriched molecules by heterotrophic bacteria would leave the residual
OM with more negative isotopic signatures (Sect. 5.2.2.). Moreover, degradation by heterotrophic bacteria leaves
more recalcitrant DOM in the water column which tends to accumulate over longer periods of time (Ogawa et al.,
2001; Jiao et al., 2010; Kawasaki et al., 2013). DOM content is a balance between its production by autotrophs
and consumption by heterotrophs, especially in environments where both types of organisms compete for nutrients
at a low content (Dittmar, 2015). If Alchichica's DOC actually represents a long-term reservoir, its presence might
favor the development of bacterial populations growing on it. Alcocer et al. (2014) describe the shift of the
cyanobacterial DOC towards the hypolimnion of Lake Alchichica at the end of the spring. Deeper and darker
anoxic waters in Alchichica could better preserve DOM from intense microbial and light degradation, hence
allowing its accumulation.
In conclusion, Alchichica's DOC reservoir (and notably in the hypolimnion) more likely represents an older and
evolved DOM pool. The time required for its accumulation and its stability over the years remain to be
investigated. Nevertheless, we cannot fully rule out that part of it this DOC was produced by anoxygenic
photosynthetic plankton. If so, the reasons why it did not bear the same isotopic enrichment as in the other lakes
also remain to be elucidated.

**5.3.2.  DOC analysis provides deeper insights into planktonic cells functioning and water column C cycle**
**dynamics**
With concentrations ranging from 0.6 and 6.5 mM on average, DOC amounts between 14 and 160 times the POC
concentrations. It represents from about 5 to 16% of total C measured in the four lakes. In comparison, although
DOC is the main organic pool in the ocean, its concentration hardly exceeds 0.08 mM (Hansell, 2013) while in
large scale anoxic basin such as the Black Sea, it remains under 0.3 mM (Ducklow et al., 2007). Hence DOC is a
major C reservoir in these Mexican lakes.
The depth profiles of DOC concentration and isotopic composition differ significantly from those of POC. Notably
in Lake La Preciosa, the photosynthetic DOC production (+1.5 mM) at the Chl. a peak depth matches the decrease
of DIC (- 2 mM) (Fig. 3) with no change in [POC] or $\delta^{13}C_{POC}$. Just below, at a 15 m depth, the marked increase of
$\delta^{13}C_{POC}$ related with heterotrophic activity (Sect. 0) might be better understood when considering the heavier DOC
isotopes compositions as a C source between 12.5 and 15 m depth (Fig. 4). In Lake La Alberca, only a small
portion of C is transferred from the inorganic to the POC by primary productivity, while the DIC reservoir is
largely dominated by methanogenesis and possible volcanic degassing in the bottom of the lake. In Lake Atexcac,
anoxygenic photosynthesis clearly stands out based on [DOC] and $\delta^{13}C_{DOC}$ data, but is not recorded by the POC
reservoir and only slightly by the DIC reservoir. Overall, it implies that recently fixed OC is quickly released out
of the cells as DOM, thereby transferring most of C from DIC to DOC, rather than POC which, therefore, is an
incomplete archive of the biogeochemical reactions occurring in water columns. Furthermore, this shows that the
isotopic analysis of DIC and by extension authigenic carbonates, especially in alkaline-buffered waters, might not
be sensitive enough to faithfully archive environmental and biological changes.
The heavy $\delta^{13}C_{DOC}$ recorded in lakes La Preciosa, La Alberca and Atexcac provides important constraints on the
way planktonic cells deal with and cycle C: it may arise from the activation of a DIC-CM or from a specific
metabolism or C fixation pathway. By contrast, the use of a DIC-CM is poorly captured by $\delta^{13}C_{POC}$ analyses,
although recognition of active DIC uptake has often been based on this signal (by reduced isotopic fractionation
with the DIC; e.g. Beardall et al., 1982; Erez et al., 1998; Riebesell et al., 2000). Most interestingly, intra-cellular
amorphous Ca-carbonates (iACC) are formed in some of the cyanobacteria from Alchichica microbialites, possibly
due to supersaturated intra-cell media following active DIC uptake through a DIC-CM (Couradeau et al., 2012;
Benzerara et al., 2014). While this link is still debated (Benzerara et al., 2014), the active use of DIC-CMs in the
studied Mexican lakes is independently supported by the DOC isotopic signature.
The report that DOC is a major C reservoir in lakes has several other implications. First, the fact that a major
fraction of primary and secondary productivity may be released and cycled as DOM instead of POM contrasts
with the conventional view that autochtonous SOM strictly records water columns biological processes. Then, if
a larger fraction of DOC incorporated the POM (e.g. due to higher nutrient availability), which later deposits as
SOM, it may tend to shift both POM and SOM isotopic compositions towards higher values (e.g. Fig. 7b *vs* 7c).
However, we notice that $\delta^{13}C_{SOC}$ does not seem to keep track of peculiar DOC isotopic signatures, although OC
carbon of the lakes is by far dominated by DOC over POC. Finally, in lakes such as Lake La Alberca, where
alkalinity is not high enough to have a high buffering effect, production or consumption of DOC should increase
or decrease, respectively, the $\delta^{13}C$ of the residual lake DIC and ultimately the isotopic signatures of authigenic
carbonates accumulated in the sediments (see below).

### 5.3.3. Implications for the inference of past big DOC reservoirs

The studied Mexican lakes have large DOC pools, allowing to draw comparisons with studies that have invoked
past occurrences of oceanic carbon cycles dominated by big DOC reservoirs (e.g. Rothman et al., 2003; Sexton et
al., 2011). Ventilation/oxidation cycles of a large deep ocean DOC reservoir have been inferred to explain
carbonate isotopic records of successive warming events through the Eocene (Sexton et al., 2011). Briefly, the
release of carbon dioxide into the ocean/atmosphere system following DOC oxidation would generate both the
precipitation of low $\delta^{13}C$ carbonates and an increase of the atmospheric greenhouse gas content. It was assessed
that the size of this DOC reservoir should have been at least 1600 PgC (about twice the size of the modern ocean
DOC reservoir) to account for a 2-4°C increase of deep ocean temperatures (Sexton et al., 2011). However, the
main counter argument to this hypothesis is that the buildup of such a DOC reservoir at modern DOC production
rates implies a sustained deep ocean anoxia over hundreds of thousand years, while independent geochemical
proxies do not suggest such a sustained anoxia during this time interval (Rigwell and Arndt, 2015). However, our
study suggests that this counter argument may be weak. Indeed, in the studied Mexican lakes, the lowest recorded
[DOC] is 260 µM (Table 1), i.e., about 6 times the deep modern ocean concentrations (~ 45 µM; Hansell, 2013).
Yet, the entire water columns of these lakes down to the surficial sediments are seasonally mixed with oxygen



showing that high [DOC] (notably in Alchichica which likely harbor a "long-term" DOC reservoir) can be
achieved despite frequent oxidative (oxygen-rich) conditions. Besides, the oxidation of only half of the DOC in
the studied lakes would generate average $\delta^{13}C_{DIC}$ deviations between -0.6 and -1 ‰, corresponding to the C
isotopes excursion magnitudes described by Sexton et al. (2011).
Similarly, Black Sea's deep anoxic waters hold about 3 times the amount of DOC found in the modern deep open
ocean (Sexton et al., 2011; Dittmar; 2015). In the Black Sea and Mexican lakes, the low nutrient availability may
limit sulfate-reduction despite high sulfate and labile organic matter concentrations, thence favoring DOM
preservation and accumulation (Dittmar, 2015 and references therein). Margolin et al. (2016) argued that important
DOM was only sustained by important terrigenous inputs. Our study attests the possibility for "autochthonous
systems" to reach DOC concentrations well above what is found in the Black Sea and that terrigenous inputs are
not needed for that. Therefore, it can be argued that the buildup of a large DOC reservoir which may have
influenced the carbonates isotopic record of Eocene warming events is plausible.
The presence of a large oceanic DOC reservoir has also been used to account for the Neoproterozoic C isotopic
record, where carbonates show $\delta^{13}C$ negative excursions of more than 10 ‰ over tens of Ma, while paired
sedimentary organic carbon isotope signal remain stable (Rothman et al., 2003; Fike et al., 2006; Swanson-Hysell
et al., 2010; Tziperman et al., 2011). However, once again, this hypothesis has been questioned because of the too
high DOC reservoir's size (10 times the contemporaneous DIC, i.e., $10^2$ to $10^3$ times that of modern DOC) and
amount of oxidants required to generate such a sustained DOC oxidation excursion (see Ridgwell and Arndt,
2015). Modeling approaches have both supported or contradicted this hypothesis: some suggested that partial
oxidation of a large DOC reservoir would suffice to explain such excursions (Shi et al., 2017), while others
concluded that DOC abundance in the past Earth's oceans could not have significantly departed from today's
values (Fakhraee et al., 2021). Critically, although multiple studies have built on the Neoproterozoic big DOC
scenario (e.g. Li et al., 2017; Canadas et al., 2022), there is at the moment no evidence – to the best of our
knowledge – for the existence of such high oceanic DOC levels in the past or present days. Modern analogous
systems such as the Black Sea or Mexican lakes studied here support the possibility of important DOC contents
accumulation but those remain substantially lower than the levels required to account for the Neoproterozoic events
(Ducklow et al., 2007; Ridgwell and Arndt, 2015).
In the studied lakes, a full DOC oxidation would generate a maximum $\delta^{13}C_{DIC}$ deviation of -2 ‰, in Alberca de
los Espinos, which has the lowest alkalinity, and the lowest $\delta^{13}C_{DIC}$. The other lakes $\delta^{13}C_{DIC}$ are less impacted,
notably because they are largely buffered by high DIC content (Table 1). Bade et al. (2004) showed that low
alkalinity/low pH lakes generally show more negative $\delta^{13}C_{DIC}$ (down to ~ -30‰), partly due to a higher response
to remineralization of OM and especially DOC. Compiling our data with those of Bade et al. (2004) we consistently
show a clear negative trend of $\delta^{13}C_{DIC}$ with increasing DOC:DIC ratio over a broad range of lacustrine DOC and
DIC concentrations (Fig. S3a). This observation is consistent with the inference that systems where DOC:DIC >>
1 should drive $\delta^{13}C_{DIC}$ to very negative values (Rothman et al., 2003). In high DOC:DIC environments, the biomass
is largely influenced by heterotrophs and usually lean towards acidic pHs (Fig. S3b; Bade et al., 2004). Hence,
environmental conditions where DOC:DIC >> 1 might be inconsistent with large carbonate deposits. Accordingly,
in light of the present results, Neoproterozoic carbonate carbon isotope excursions seem unlikely to be explained
by the big DOC scenario, unless DOC and DIC pools are spatially decoupled (e.g. through terrestrial DOM inputs).



## 6. CONCLUSIONS AND SUMMARY

The carbon cycle of four stratified alkaline crater lakes were described and extensively compared, including the concentration and isotopic compositions of DIC, DOC, POC and surficial (~10 cm) sedimentary carbonates and organic carbon (SOC) in parallel with their physico-chemical characteristics. We identify different regimes of C cycling in the four lakes due to different biogeochemical reactions related to slight environmental and ecological changes. In more details, we show that:

- external abiotic factors such as the hydrological regime and the inorganic C sources to the lakes control their alkalinity and thus, the buffering capacities of their waters. In turn, it constrains the stratification of the water columns and thus the distribution of microbial communities and their respective metabolic effect on C.

- Based on POC and DIC concentrations and isotopic compositions, coupled with physico-chemical parameters, we are able to identify the activity of oxygenic photosynthesis and aerobic respiration in the four studied lakes. Anoxygenic photosynthesis and/or chemoautotrophy as well as sediment-related methanogenesis are also evidenced in some of the lakes, but their POC and DIC signatures can be equivocal.

- DOC is the largest OC reservoir in the water column of the studied lakes (> 90%). Its concentrations and isotopic compositions bring precious new and complementary information about the C cycle of these stratified water bodies. Depending on environmental factors such as nutrients and DIC availability, diverse photosynthetic planktonic communities appear to release more or less important amounts of DOC depending on the lake, transferring most of the inorganic C fixed to DOC rather than POC. This process is marked by very heavy and distinct isotopic signatures of DOC compared to POC. They reflect different metabolism/C fixation pathways and/or the activity of a DIC-CM coupled with an overflow mechanism (i.e. DOM exudation) for which we propose a novel isotopic model including DOC. These features are invisible to POC analyses and thus are not recorded in the sediments.

- Our results bring further constraints on the environmental conditions in which autochthonous DOM can accumulate in anoxic water bodies and provides boundary conditions to the "big DOC reservoir" scenario.

- We observe that the SOM geochemical signatures of these stratified lakes do not all record the same biogeochemical layers of the water column and can be largely modified by early diagenesis in some cases.

- Methanogenesis is evidenced in the surficial sediments of the organic-rich Lake La Alberca de los Espinos and influences the lower water column geochemical signatures. However, it is recorded only by analyses of pore water dissolved species and not in its sedimentary archives (OM and carbonates).

**Author Contributions**
RH and CT designed the study in a project directed by PLG, KB and CT. CT, MI, DJ, DM, RT, PLG and KB collected the samples on the field. RH carried out the measurements for C data; DJ the physico-chemical parameter probe measurements and EM provided data for trace and major elements. RH and CT analyzed the data. RH wrote the manuscript with important contributions of all co-authors.



**Competing Interests**
The authors declare that they have no conflict of interest.

**Disclaimer**

**Acknowledgements**
This work was supported by Agence Nationale de la Recherche (France; ANR Microbialites, grant number ANR-
18-CE02-0013-02). The authors thank Anne-Lise Santoni, Elodie Cognard, Théophile Cocquerez and the GISMO
platform (Biogéosciences, University Bourgogne Franche-Comté, UMR CNRS 6282, France). We thank Céline
Liorzou and Bleuenn Guéguen for the analyses at the Pôle Spectrométrie Océan (Laboratoire Géo-Océan, Brest,
France) and Laure Cordier for ion chromatography analyses at IPGP (France). We thank Nelly Assayag and Pierre
Cadeau for their help on the AP 2003 at IPGP.

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
