# Peer review of "A comparative isotopic study of the biogeochemical cycle of carbon in modern redox-stratified lakes"

_Biogeosciences, 2022_

## Author Response (AR1)

**Authors' response**

The reviewers suggested a major revision of our initial manuscript "A comparative isotopic study of the biogeochemical cycle of carbon in modern stratified lakes: the hidden role of DOC" and notably recommended to split this study in two different manuscripts.

Following their recommendation, we split the manuscript in two and followed an outline that was presented in detail in our public answers to the reviewers (available in the interactive discussion dedicated to our study) and validated by the associate editor.

Following this outline, we now submit the first of these two manuscripts entitled: "A comparative isotopic study of the biogeochemical cycle of carbon in modern redox-stratified lakes". In brief, it will present and discuss the DIC and POC results of the initial manuscript, while the second manuscript ("The hidden role of DOC in the biogeochemical cycle of carbon in modern redox-stratified lakes") will tackle the DOC dataset. Both will be submitted to Biogeosciences journal.

As a consequence of this major reworking, the first reviewer did not provide a detailed review of our work and thus, we do not report here a list of specific answers. Nevertheless, we took all the reviewers' comments into consideration during the reformatting processes.

We considered the comment made by the second reviewer on the introduction section (and responded in the online public reply) and modified the corresponding text accordingly. In the "author's track-changes file", we addressed and answered the minor comments brought up by the reviewer and which will belong to the presently submitted revision of the first manuscript.

Since there was no major disagreement from the reviewers with our datasets nor their interpretation, there was no major change made to the flow of argumentation within the discussion parts. However, we slightly reorganized the discussion sections according to the reviewers' suggestion, that is, centered around the sources and sinks description of DIC and POC reservoirs. Additionally, in a way to make the manuscript as a stand-alone piece of work, we discuss in more details than in the previous version the sinks of DIC – and notably the precipitation of sedimentary carbonates and their C isotopic compositions.

Overall, we think that the modifications applied to this new version markedly improve its consistency and clarity, and thus easier to catch the important messages from. We hope the reviewers and the editorial board will also see the benefits of these changes.

Yours sincerely,
      Robin Havas, on behalf of all co-authors

Answers to the comments made by reviewer 2 (as there was no detailed review of the manuscript by reviewer 1) and belonging to the first resubmitted paper.

*Line 161: What semi-calibrated means?* We clarified this sentence: "The ORP signal was not calibrated before each profile and is thus used to discuss relative variations over a depth profile." (Line 145, new manuscript)

*Line 165: Why different volumes?* Different volumes were filtered depending on the volume that would reach clogging of the filters. Now explained in the text (Lines 149-150).

*Line 166: Does pre-ashed mean pre-combusted? Add temperature and time if this is the case.* Done, and changed pre-ashed to pre-combusted (Line 148).

*Line 171: Please provide details on the transport from the lake to the laboratory.* Done (Lines 157-158).

*Line 175: Was the water filtered, stored in a container and then, put in the Exetainer? Or the filter went directly into the exetainer?* The water was first filtered at 0.7 µm and poured into a container. Then, the filtered-water was filtered again at 0.2 µm and poured directly into an exetainer. Now clarified in the text lines 171-172.

*Line 197: The name of the laboratory should be written in either French or English throughout the manuscript, and the city in which the laboratory is located should be mentionned. Currently we have a mix of French and English.* Modified throughout the manuscript.

*Line 265: Place into brackets and move "between 34.5 and 35 mM" after "throughout the water column".* This sentence was rearranged (Line 248).

---

## Referee Report (RR1)

The manuscript by Havas et al. (A comparative isotopic study of the biogeochemical cycle of carbon in modern redox-stratified lakes) provides a very detailed dataset of water column and near-surface sediments from four small lakes in Mexico. Three lakes are located in the Serdan-Oriental Basin, composed mainly of limestone, and one lake is further west is in the Michoacán-Guanajuato Volcanic field, which is mainly composed of andesitic rocks. The contrasting lithologies provide different sediment input into the lakes.

The manuscript is part of a much larger study that was previously submitted to Biogeosciences. I considered the previous manuscript to be a whale of a paper, so I was happy to see it cut into more palatable pieces. At least now it gives the readers a chance to follow through the story, but the present manuscript is still massive!

This is an excellent dataset and I don't have much to criticize on the data, except perhaps the missing error bars on the POC values in fig. 3.

General comments
- The language is acceptable, but sometimes I get the impression that there is still a bit of "Frenglish" in the text and that the word "the" is used somewhat randomly. Also, the text is sometimes going back and forth between past and present. Another issue is singular/plural and the use of the possessive "s" or the apostrophe in lieu of the letter. I would recommend a thorough revision by a professional language editing service or a colleague who is a native speaker.
- The discussion is divided into several parts, each dealing with a specific sub-topic. The parts are well written, although I sometimes felt that the authors got a bit carried away by the many details. I would recommend to take a hard look at the discussion and ask whether every detail is actually necessary to get the general story across.
- What I really liked were the little summaries at the end of some sub-chapters, e.g. 5.1.3 and 5.1.4. In each of the sub-chapters the final paragraph sums up the story. I would appreciate if the authors could add these little summaries to every part of the discussion so the readers get a clear take-home message.
- Quite a few datasets are presented both in figure and table, e.g. fig 5 and table 5. This is redundant. It is nice to have the actual numbers, but please put them into the supplement and just leave the figures in the text. Overall, I would avoid presenting data in tables in the main text. Tables are fine for the SI, so the readers can look up the actual numbers if they really need to, in the main text just use figures.

Specific comments
Line 31f: What do you mean by "varied correlatively"? Please rephrase the sentence
Line 36: …we identified…
Line 80: …allows for assessing the effects…
Line 81: What do you mean by "correspond to"?
Line 82: ….allows discussing their influence….
Line 85: ..on the lakes' stratification…..
Line 129: ….but cover a wide range of chemical compositions….
Line 131: …in concentration stages….
Line 153: What influence does the temporary bottom water anoxia have on the lakes' systems? This should also be discussed in the discussion.
Figure 1: Please change the order of the photos to avoid the lines crossing over. From left to right: Alberga de los Espinos, Atexac, Alchichica, La Preciosa

Line 191: How did you dry the samples before grinding?

Figure 2: How can the ORP remain stable when oxygen becomes depleted? In all 4 lakes DO runs out at some depth and ORP remains absolutely stable down to much greater depths before it starts decreasing as well? I've worked on many lakes but usually ORP more or less follows DO, so please add an explanation.

line 262: Conductivity showed the same trend with values…

Table 2: Please move to SI, present data in figure in main text

Line 370: How does the groundwater flow? Your explanation is hard to follow. Could you add a sketch of the flow paths?

Table 3: Put into SI and make figure for main text

Line 422: What do you mean by …one hand, those from…. I don't know what the "those" means

line 436: …is lower than….

line 455: is the offset of a few permill actually relevant and/or is the accuracy of the measurements sufficient to detect this?

Line 461: What do you mean by …relatively important storage….?

Line 469: Please add reference for the statement….degassing through higher pCO2 (despite high pH values).

Line 474: another important sink of CO"…you are talking about sources of CO2 before, somehow I am missing the connection here

Line 475: …microbialites and lake sediments. Please refrain from writing bottom lake sediments. Sediments are always at the bottom unless they are (re)suspended.

Line 475: ….alkalinities and resulting mineral saturation greatly influence….

Line 486: Why is the lake a sink for CO2 when its surface waters are in equilibrium?

Table 4: Move to SI

Line 524:…POM that ranged from 6 to 12 in….

Line 531: …source of POC in the four lakes…

Figure 5, Table 5: This is redundant, same information

Line 614: Please add reference for your statement about sulfur-oxidizing bacteria

Line 665: Is Methane loss through degassing to the atmosphere realistic in these lakes? I don't think so but I might be wrong.

Line 709: …size of phytoplankton… Also, what do you mean by large size of phytoplankton?

Lines 710/11: No! Please see Friese et al. (2020, Nature Communications, https://doi.org/10.1038/s41467-021-22453-0) or Vuillemin et al. 2016 (Frontiers in Microbiology, DOI: 10.3389/fmicb.2016.01007) and other literature about ferruginous lakes. It clearly shows that even at such low sulfate concentrations, sulfate reduction can proceed at appreciable rates due to reoxidation of reduced sulfur species.

Line 733: …lakes does not…..

line 736: Move "instead" to the end of the sentence

Line 737: …was highest in….

line 738: …from water column to sediment will…..

Lines 748-750: Please rephrase that sentence

---

## Author Response (AR2)

In this document, we present our response to the two reviews that were provided on our manuscript initially entitled "**A comparative isotopic study of the biogeochemical cycle of carbon in modern redox-stratified lakes**". We first would like to thank both reviewers for accepting this task and providing helpful comments in the aim of improving our manuscript.

The comments of the reviewers are written in black and the line numbers they indicated were unchanged (that is, these line numbers correspond to those in the initially submitted manuscript). We answered to each of the reviewers' comments in blue, and the line numbers we indicate correspond to the line numbers in the new modified manuscript.

**Review N: 1 (accepted with minor corrections, will not read the manuscript again)**

The manuscript by Havas et al. (A comparative isotopic study of the biogeochemical cycle of carbon in modern redox-stratified lakes) provides a very detailed dataset of water column and near-surface sediments from four small lakes in Mexico. Three lakes are located in the Serdan-Oriental Basin, composed mainly of limestone, and one lake is further west is in the Michoacán-Guanajuato Volcanic field, which is mainly composed of andesitic rocks. The contrasting lithologies provide different sediment input into the lakes.

The manuscript is part of a much larger study that was previously submitted to Biogeosciences. I considered the previous manuscript to be a whale of a paper, so I was happy to see it cut into more palatable pieces. At least now it gives the readers a chance to follow through the story, but the present manuscript is still massive!

This is an excellent dataset and I don't have much to criticize on the data, except perhaps the missing error bars on the POC values in fig. 3.

We thank the reviewer for this positive appreciation of our manuscript. We are very grateful for the reviewer's pertinent remarks on the science, as well as the detailed comments on language use, which have been very helpful.

We have added error bars in Fig. 3. For lakes La Alberca and La Preciosa, the error bars are included within the data points.

General comments
• The language is acceptable, but sometimes I get the impression that there is still a bit of "Frenglish" in the text and that the word "the" is used somewhat randomly. Also, the text is sometimes going back and forth between past and present. Another issue is singular/plural and the use of the possessive "s" or the apostrophe in lieu of the letter. I would recommend a thorough revision by a professional language editing service or a colleague who is a native speaker.

The text was reread by a colleague who is a native speaker of British English to correct the problematic grammar and phrasing.

• The discussion is divided into several parts, each dealing with a specific sub-topic. The parts are well written, although I sometimes felt that the authors got a bit carried away by the many details. I would recommend to take a hard look at the discussion and ask whether every detail is actually necessary to get the general story across.

Some aspects of the discussion may indeed have been slightly over-described. We have therefore deleted several sentences and over-precise details throughout the text, while preserving the consistency of the arguments. The short summaries added at the end of each sub-section (as recommended in the next comment) should guide the reader through this long manuscript.

• What I really liked were the little summaries at the end of some sub-chapters, e.g. 5.1.3 and 5.1.4. In each of the sub-chapters the final paragraph sums up the story. I would appreciate if the authors could add these little summaries to every part of the discussion so the readers get a clear take-home message.

We agree that these summaries will be very helpful for the readers. Thus, we have added or adapted the summaries where needed (at the end of sections 5.1.1, 5.1.2, 5.2.1., and 5.2.2).

• Quite a few datasets are presented both in figure and table, e.g. fig 5 and table 5. This is redundant. It is nice to have the actual numbers, but please put them into the supplement and just leave the figures in the text. Overall, I would avoid presenting data in tables in the main text. Tables are fine for the SI, so the readers can look up the actual numbers if they really need to, in the main text just use figures.

Following the reviewer's suggestion, we have moved the redundant tables to the SI (and deleted Table 5 as the raw data needed for these calculations are already presented in Table S1 – following the new numeration).

Specific comments
Line 31f: What do you mean by "varied correlatively"? Please rephrase the sentence

We meant that both the alkalinity and $\delta^{13}C_{DIC}/\delta^{13}C_{Carb}$ in the lakes were correlated and increased from Lake La Alberca to Lake Alchichica. The abstract has been entirely rewritten in order to make it clearer as a whole, and this passage is no longer in the text.

Line 36: …we identified…

We have modified this line and homogenized the use of present and past tenses throughout the text.

Line 80: …allows for assessing the effects… Modified in the text.
Line 81: What do you mean by "correspond to"?

We mean that they **are** closed lakes, now clarified in the text (Line 86).

Line 82: ….allows discussing their influence…. Modified in the text.
Line 85: ..on the lakes' stratification….. Modified in the text.

Line 129: ….but cover a wide range of chemical compositions….

The mention of a gradient referred to the work of Zeyen et al. (2021) and to the idea that there is a specific order in the increase of alkalinity, Mg:Ca, etc. among the studied lakes. For more clarity, without losing the meaning of the sentence, we have modified the text according to your suggestion (Line 133).

Line 131: …in concentration stages…. Modified in the text.
Line 153: What influence does the temporary bottom water anoxia have on the lakes' systems? This should also be discussed in the discussion.

We agree and we now discuss this aspect in lines 676-678.

"While the yearly mixing oxidizes most of the water column during the winter, it also generates a bloom of diatoms which fosters OM production (through shuttling up of bio essential nutrients such as N and Si) and development of anoxia (*e.g.* Adame et al., 2008)."

Figure 1: Please change the order of the photos to avoid the lines crossing over. From left to right: Alberga de los Espinos, Atexac, Alchichica, La Preciosa

We agree that lines crossing over is not ideal in terms of esthetics and we have moved the lines in order to avoid that problem. The best order for the photos is that which follows the gradient of alkalinity. The lakes are therefore presented in the following order: La Alberca de los Espinos, La Preciosa, Atexcac, and Alchichica. To reinforce this idea, we have added an arrow indicating the alkalinity gradient. Also, in order to be more consistent with this gradient, we have reorganized the results section in the same order (i.e. from La Alberca to Alchichica).

Line 191: How did you dry the samples before grinding?

The filters were dried at room temperature in petri dishes. Now specified in the text (Lines 195).

Figure 2: How can the ORP remain stable when oxygen becomes depleted? In all 4 lakes DO runs out at some depth and ORP remains absolutely stable down to much greater depths before it starts decreasing as well? I've worked on many lakes but usually ORP more or less follows DO, so please add an explanation.

This was discussed in the previous version of the manuscript but was deleted in order to shorten it and make it more concise. Your remark is absolutely correct and the observation is surprising at first. In lakes La Preciosa and Atexcac (< 2m), the delay between DO exhaustion and ORP decline is limited, while in La Alberca and Alchichica, it is indeed very significant (7 to 10 m). Two possibilities could explain this discrepancy. First, we notice that the ORP usually starts decreasing after a turbidity peak appears, likely corresponding to Mn-oxide precipitation (cf. Fig. 2). It is possible that the ORP signal is buffered to high values by the presence/formation of such oxidized species, despite the absence of $O_2$, until the probe encounters important dissolved reduced species. Second, in lakes La Alberca, La Preciosa, Alchichica, the decrease in ORP is more closely associated with the end of Chl. a and/or phycocyanin peaks. This suggests that there may still be a local production of DO, but it is quickly reduced and thus not measured. It buffers the ORP signal until Chl a and/or phycocyanin disappear. We summarized this explanation in the results section of La Alberca and Alchichica and added the above details in the supplementary material.

line 262: Conductivity showed the same trend with values… Modified in the text. (Line 297).
Table 2: Please move to SI, present data in figure in main text

We have moved Table 2 to SI and the data are now presented briefly in the main text and in Figure 4.

Line 370: How does the groundwater flow? Your explanation is hard to follow. Could you add a sketch of the flow paths?

For a better visualization of groundwater flow, we have added the main flow paths as white arrows in Fig. 1.

Table 3: Put into SI and make figure for main text

Table 3 was moved to the SI as Table S3. The parameters that show noticeable variation with depth (C:N$_{SOM}$, $\delta^{13}C_{SOC}$,[DIC] and $\delta^{13}C_{DIC}$) are presented in figures 3 and 4, whereas the [SOC], [carbonates] and $\delta^{13}C_{Carb}$ are only discussed in terms of mean values, which are reported in the text.

Line 422: What do you mean by …one hand, those from…. I don't know what the "those" means

We have clarified this by replacing "those" with "the $\delta^{13}C_{DIC}$" (Line 443).

line 436: …is lower than…. Modified in the text. (Line 454).

line 455: is the offset of a few permill actually relevant and/or is the accuracy of the measurements sufficient to detect this?

Indeed, considering the error associated with the measurement of $\delta^{13}C_{DIC}$, we agree that it is difficult to firmly conclude about isotopic disequilibrium between the lake DIC and carbonates $\delta^{13}C_{DIC}$ in Atexac. Therefore, we have deleted the sentences about putative detrital inputs and modified the text as follows: "Surficial sedimentary carbonates are in isotopic equilibrium with the $\delta^{13}C_{DIC}$ of the water columns, within the uncertainty of $\delta^{13}C_{DIC}$ measurement, and more specifically with the $\delta^{13}C_{DIC}$ values at the oxycline/thermocline of the lakes (Tables S6 and S7)." (Lines 462-464)

Line 461: What do you mean by …relatively important storage….?

We mean that high pH allows for a higher quantity of DIC than low pH. To clarify, we have modified the text as follows: "Alkaline pH can store large quantities of DIC because…" (Lines 416).

Line 469: Please add reference for the statement….degassing through higher pCO2 (despite high pH values).

We have added a reference here. To simplify, we also deleted "(despite high pH values)" as it may cause confusion about the role of pH in $CO_2$ degassing (Lines 422).

Line 474: another important sink of CO"…you are talking about sources of CO2 before, somehow I am missing the connection here

We agree that it may appear confusing here. Considering the lakes' DIC reservoir, $CO_2$ degassing discussed just before represents a C sink, since $CO_2$ is lost to the atmosphere. To clarify, we have deleted the sentence about Alchichica, Atexcac and La Preciosa acting as $CO_2$ sources to the atmosphere in the previous paragraph.

Line 475: …microbialites and lake sediments. Please refrain from writing bottom lake sediments. Sediments are always at the bottom unless they are (re)suspended.

Ok, we have deleted "bottom" here and throughout the text, except for the first occurrence.

Line 475: ….alkalinities and resulting mineral saturation greatly influence…. Modified in the text (Line 427).

Line 486: Why is the lake a sink for CO2 when its surface waters are in equilibrium?

We agree with the reviewer that this sentence is incorrect because the pCO2 of this lake was actually lower than that of the atmosphere, explaining why it represented a C sink at the time of sampling. Modified line 434.

Table 4: Move to SI Ok.

Line 524:…POM that ranged from 6 to 12 in…. Modified in the text (Line 497).

Line 531: …source of POC in the four lakes… Modified in the text (Line 503).

Figure 5, Table 5: This is redundant, same information

We have deleted Table 5 to shorten the manuscript.

Line 614: Please add reference for your statement about sulfur-oxidizing bacteria

In the reference, van Vliet et al. (2021) actually also discuss the oxidation of sulfur compounds by sulfur-oxidizing bacteria. To avoid the addition of new references, we moved this reference to the end of the sentence.

Line 665: Is Methane loss through degassing to the atmosphere realistic in these lakes? I don't think so but I might be wrong.

We agree that methane is likely not escaping from Lake Alberca as we suggest in the text, but this does occur in other similar modern lakes (see e.g. Cadeau et al., 2020). Thus, we discuss this possibility in that part of the discussion, along with that of methane being oxidized in the water column.

Line 709: …size of phytoplankton… Also, what do you mean by large size of phytoplankton?
We refer to sizes of organisms of several μm to several tens of μm (see Adame et al., 2008 and Ardiles et al., 2011).

Lines 710/11: No! Please see Friese et al. (2020, Nature Communications, https://doi.org/10.1038/s41467-021-22453-0) or Vuillemin et al. 2016 (Frontiers in Microbiology, DOI: 10.3389/fmicb.2016.01007) and other literature about ferruginous lakes. It clearly shows that even at such low sulfate concentrations, sulfate reduction can proceed at appreciable rates due to reoxidation of reduced sulfur species.

We understand the reviewer's comment and we agree that some degree of sulfate reduction is active in Lake La Alberca despite low $SO_4$ concentrations. Indeed, as stated in the text and seen in figures 2 and S5 of our manuscript, $SO_4$-reduction is evidenced by pyrite precipitation in the bottom water column and sediments. Thus, we do not imply that no sulfate reduction takes place, but that limited $[SO_4]$ favors OM preservation. Although Vuillemin et al. (2016) underline the importance of rapid S cycling allowing appreciable $SO_4$-reduction to occur despite μM-level $[SO_4]$, they do mention that the lower the $[SO_4]$, the lower the $SO_4$-reduction rate. Moreover, after describing the lower and upper limits of sulfate reduction rates, Friese et al. (2020) state that "…like iron reduction, sulfate reduction plays only a minor role in organic matter degradation" and later in the text that it "makes a minor, even insignificant, contribution to total organic carbon mineralization". Similarly, other studies have shown that most of the organic matter from ferruginous/$SO_4$-poor Matano lake was buried and preserved in the sediments (e.g. Kuntz et al., 2015).

In order to clarify our point in this paragraph we have modified the text as follows: " Because bacterial sulfate reduction (BSR) is a major remineralization pathway in $SO_4$-rich environments (e.g Jørgensen, 1982). the low sulfate content in La Alberca probably favors the preservation of high TOC in the sediments. Even though, appreciable BSR rates may occur in this lake (see discussion above and Fig. S5), similarly to other sulfate-poor environments due to rapid S-cycling (e.g. Vuillemin et al., 2016; Friese et al., 2020)" (Lines 679-683).

Line 733: …lakes does not….. Done.
line 736: Move "instead" to the end of the sentence Done.
Line 737: …was highest in…. Modified.
line 738: …from water column to sediment will….. Modified.
Lines 748-750: Please rephrase that sentence

We have rephrased the sentence as follows: "In turn, these different buffering capacities constrain the variations of pH along the stratified water columns as well as the inorganic C isotope signatures recorded in the water columns and sediments of the lakes". (Lines 720-722).

**Review N: 2 (reconsidered after major revision, will be willing to review the revised manuscript)**

In this revision, the authors reduced the length of the manuscript and better focused their ideas. Whereas the efforts are appreciated, I unfortunately still have the same major concern as in the first round of revision. That is, the authors still need to convey their ideas more concisely. The dataset is nice, I think, but it is very hard to understand how the authors reached their conclusion. The hypothesis of the authors is too ambitious. Surely, this manuscript is not describing the entire carbon cycle, and the hypothesis line 69 needs to be toned down. I think the lack of clear hypothesis prevents the authors from delivering a manuscript that is easy to follow and understand.

We thank the reviewer for this positive appreciation of our shortened manuscript. Following the reviewer's suggestions, we have restructured the abstract and the introduction to present a clear hypothesis, and the aims of our study. The goal of our study is to investigate how environmental and biological differences are captured by the carbon isotope signatures in the water columns (DIC-POC) of four modern redox-stratified lakes, and transferred to the sedimentary archives (OM-carbonates). The aim of the paper is not to describe the entire C cycle of these lakes, and we now clarified this aspect.

The discussion section still contains too many results that were not presented in the results section.

The discussion is now organized in a clearer and more straightforward way, describing first the sources and sinks of DIC, and then of POC.

We have added systematic summary sentences at the end of sub-sections to allow the reader to better follow the text and identify the important take home-messages.

We have checked that all results discussed are presented in the results section.

Also, a substantial portion of the discussion is neither explaining the data nor describing how the data improve our knowledge of the carbon cycle.

For instance, the entire section "Influence of methanogenesis in Lake La Alberca de los Espinos" is based on assumptions, and the authors should not emphasise that they have identified the presence of methanogenesis. If this manuscript shows the presence of methanogenesis, we need to know how it was measured in the methods section. This cannot only come in the discussion. If there is no methods and no results on methanogenesis, why do we have 50 lines on it? If the authors want to talk about methanogenesis, maybe reduce that to 5-10 lines.

The discussion sections are based on the DIC/POC data and physico-chemical parameters described in the results section, to address the hypothesis presented above, i.e., how the environmental and biological characteristics of the four lakes are recorded in the inorganic and organic C reservoirs.

We identified the presence of methanogenesis in Lake La Alberca based on specific DIC/POC isotopic signatures, and so we did not present a direct method for methanogenesis identification/quantification (e.g. metagenomics). The dedicated section does, however, clearly demonstrate the presence of this OM remineralization pathway, and how it is detected by [DIC]/$\delta^{13}C_{DIC}$ but not by $\delta^{13}C_{Carb}$. Thus, in our view, the discussion on methanogenesis fits well with the objective of showing how a metabolism identified in a specific environment may not necessarily be recorded in the sedimentary archives.

Overall, I would again suggest to the authors to find a clear hypothesis and to provide a concise manuscript based on their data.

We hope that this revised version, taking into account all the reviewer's suggestions, will be found satisfactory. Once again, we thank the reviewer for these helpful comments, which have improved our paper.

---

## Author Response (AR3)

In this revision, the authors better introduce their ideas, and it is easier to understand the manuscript. I am still not convinced by the discussion as it contains too many speculations and too many subsections to my taste. There are many "most likely", "probably", and "may be" in the discussion which clearly indicates that the authors do not have empirical evidence to make the claim they would like. However, this version has a hypothesis, I did find the answer to that hypothesis in the results and discussion sections, the methods sound good to me, the data supports the conclusion, and the study has a significant importance. Therefore, my taste is simply an opinion and maybe most people will be happy to have all those elements discuss. Much like the other reviewer, I think the manuscript will read better if the authors spend some time cleaning the grammar. For instance, there are too many sentences starting with "this/these" and it is not easy to follow what we are talking about. The text is also quite wordy.

We thank the reviewer again for commenting on and helping to improve our manuscript. In total, we now have replaced or complemented 22 instances of "this/these" that were not clear enough. The manuscript was reread and corrected by a native British English speaker.

Some additional comments:
Line 28: "sedimentological expression" is not clear. The use of "these processes" is also not helping because "these processes" were not introduced before. Or at least they were not introduced as processes, so the reader has to guess what we are talking about. In general, too many sentences in the abstract start by "this/these" and it is not easy to identify the subject of those sentences.
We have rephrased this passage in the abstract. We have replaced "this/these" where the link with previously introduced objects was not clear enough.

Line 371: More concentrated in what? In salt?
We have clarified by specifying that the groundwater's "ionic strength (including DIC concentration) increases" as groundwater flows towards Alchichica.

Line 495-496: That is moving too fast as it sounds like the authors claim that because the lakes are not connected to a river, autochthonous processes dominate. I doubt the authors had such an intention.
Indeed, surface streams and rivers are expected to transport a significant portion of organic carbon inland (Hotchkiss and Hall, 2015; doi: 10.1890/14-0631.1). The absence of such streams should greatly reduce the input of allochthonous OM to the lakes studied, especially POM. We have reorganized the paragraph, referring to the lakes' endorheic nature only at the end. We have also added a reference supporting the same inference for Lake Alchichica (reference added on line 503).

Line 553-555: The authors claim that an increase in POC below the oxycline is indicative of anoxygenic primary production. Other processes, such as sedimentation could also increase the POC below the oxycline. Therefore, the claim made by the authors needs further justification. I take that there is a subsection later on talking about sedimentation. However, the claim is made before such a subsection and I think the authors, not the reader, should make the link between the data and their different interpretation.
We state in the text that the increase of POC with depth in La Alberca "suggests" the possible anoxygenic autotrophy occurring in this lake. In the following lines 558-562, we further justify this suggestion. To reinforce the claim of anoxygenic autotrophy in La Alberca, we have added that the [POC] increase was associated with a distinctive isotopic signature, as seen in figures 4 and 5, arguing against the hypothesis of sedimentation.

Line 566: "at depth". Is there a number missing here or "at depth" stands for "below the oxycline"?

We have replaced "at depth" by "below the oxycline".

Figure 3: The figure seems cropped, and we cannot see all the data.

We have modified figure 3 accordingly.